# Conserved lipid metabolic reprogramming confers hypoxic and aging resilience

Wei I Jiang [1], Goncalo Dias do Vale [2], Quentinn Pearce[3,4], Kaitlyn Kong[1], Wenbin Zhou[1], Jeffrey G McDonald [2], James E Cox[3,4], Neel S Singhal [5] & Dengke K Ma [1,6,7]✉

## Abstract

The Arctic ground squirrel (AGS, *Urocitellus parryii*), an extreme hibernator, exhibits remarkable resilience to stressors like hypoxia and hypothermia, making it an ideal model for studying cellular metabolic adaptation. The underlying mechanisms of AGS resilience are largely unknown. Here, we use lipidomic and metabolomic profiling to discover specific downregulation of triglyceride lipids and upregulation of the lipid biosynthetic precursor malonic acid in AGS neural stem cells (NSC) versus murine NSCs. Inhibiting lipid biosynthesis recapitulates hypoxic resilience of squirrel NSCs. Extending this model, we find that acute exposure to hypoxia downregulates key lipid biosynthetic enzymes in *C. elegans*, while inhibiting lipid biosynthesis reduces mitochondrial fission and facilitates hypoxic survival. Moreover, inhibiting lipid biosynthesis protects against APOE4-induced pathologies and aging trajectories in *C. elegans*. These findings suggest triglyceride downregulation as a conserved metabolic resilience mechanism, offering insights into protective strategies for neural tissues under hypoxic or ischemic conditions, APOE4-induced pathologies and aging.

**Keywords** Arctic Ground Squirrel; Triglyceride Lipids; Hypoxia; Lipid Biosynthetic Enzymes; *C. elegans*
**Subject Category** Metabolism

## Introduction

Oxygen is essential for survival in animals and humans, and its dysregulation can lead to devastating diseases (Baik and Jain, 2020; Kaelin and Ratcliffe, 2008; Semenza, 2012). For instance, cerebral ischemic stroke results from an inadequate oxygen supply (hypoxia) to the brain, leading to neuronal injury or death. Arctic ground squirrels (AGS, *Urocitellus parryii*) inhabit the extremely cold environments of the Arctic and sub-Arctic, where temperatures can drop drastically. To survive these conditions, AGS have

evolved physiological adaptations that enable them to withstand both prolonged exposure to low temperatures and periods of reduced circulation and oxygen availability during hibernation (Chmura et al, 2023; Yan et al, 2008; Williams et al, 2011; Lee and Hallenbeck, 2006; Rice et al, 2020). Notably, their ability to maintain cellular function and resist hypoxia-induced damage can be recapitulated ex vivo in brain slices and cell culture models, suggesting genetically innate or intrinsic mechanisms for stress resilience (Williams et al, 2016; Drew et al, 2016; Dave et al, 2006; Singhal et al, 2020). Such cell-intrinsic adaptations provide a powerful model to investigate cellular and molecular strategies for hypoxia tolerance. Ex vivo cell-based models derived from AGS also offer tractability to dissect these mechanisms in a controlled setting, allowing researchers to uncover pathways that may inform therapeutic strategies for stroke, heart attacks, and other conditions associated with oxygen deprivation (Bhowmick and Drew, 2019; Drew et al, 2023; Zhao et al, 2021).

The nematode *Caenorhabditis elegans* (*C. elegans*) serves as an instrumental model system for dissecting evolutionarily conserved hypoxia signaling pathways (Powell-Coffman, 2010; Epstein et al, 2001; Shen and Powell-Coffman, 2003). Genetic analysis of the *C. elegans* egg-laying behavioral mutant *egl-9* led to the identification of a conserved family of oxygen-sensing HIF hydroxylases in animals (Epstein et al, 2001). Like its mammalian counterpart, HIF-1 (hypoxia-inducible factor 1) in *C. elegans* is stabilized under hypoxic conditions owing to impaired proline hydroxylation by EGL-9 (HIF hydroxylase) and subsequent degradation by VHL-1 (Von Hippel–Lindau). Hypoxia-induced HIF-1 activation facilitates cellular adaptation and enhances survival under hypoxia. Notably, stabilization of HIF-1 through VHL-1 inhibition has been shown to protect dopamine neurons from oxidative stress, degeneration and mitigate APOE4-induced neural pathologies (Johansen et al, 2010; Chen et al, 2019; Jiang et al, 2025). While extensive research has identified numerous transcriptional targets of HIF-1 and elucidated the genetic and cellular mechanisms through which HIF-1 regulates mitochondrial metabolic functions in response to hypoxia (Kim et al, 2006; Angeles-Albores et al, 2018; Vora et al, 2022; Comas-Ghierra et al, 2023; Fukuda et al, 2007; Feng et al, 2024a, 2024b; Warnhoff et al, 2024; Doering et al, 2022; Kruempel et al, 2020; Ma et al, 2012; Pender and Horvitz,

[1]Cardiovascular Research Institute, University of California San Francisco, San Francisco, CA, USA. [2]Center for Human Nutrition and Department of Molecular Genetics, University of Texas Southwestern Medical Center, Dallas, USA. [3]Metabolomics Core Research Facility, University of Utah, Salt Lake City, UT, USA. [4]Department of Biochemistry, University of Utah School of Medicine, Salt Lake City, UT, USA. [5]Department of Neurology, University of California San Francisco, San Francisco, USA. [6]Department of Physiology, University of California San Francisco, San Francisco, CA, USA. [7]Innovative Genomics Institute, Berkeley, CA, USA. ✉E-mail: Dengke.Ma@ucsf.edu

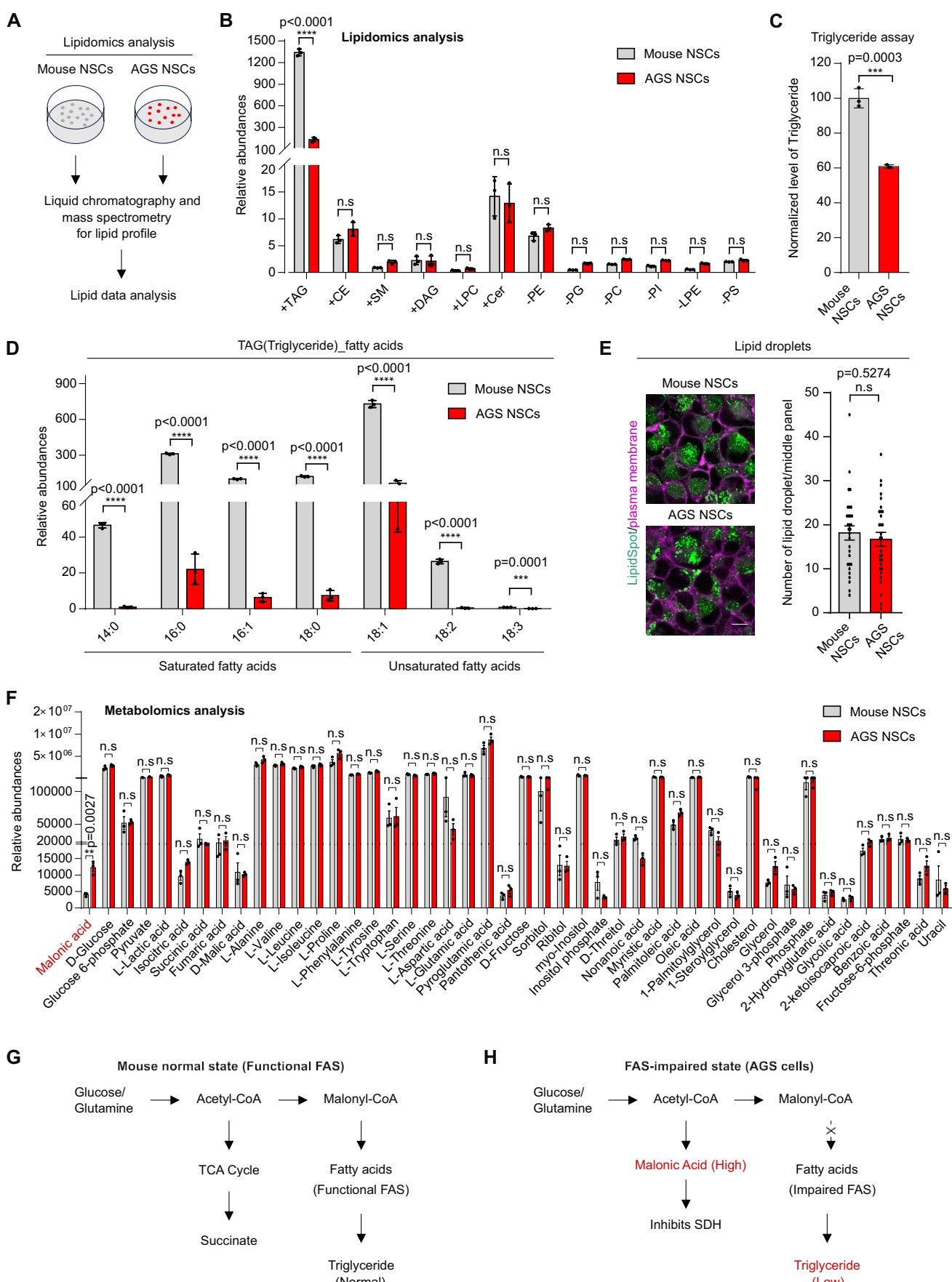

**Figure 1.  Saturated and unsaturated triglycerides are downregulated in AGS.**

(A) Schematic of the lipidomic analysis of mouse neural stem cells (NSCs) and Arctic ground squirrel NSCs under normal cell culture conditions. (B) Quantification of lipids in AGS NSCs and mouse NSCs showing that triglycerides (normalized to protein) were significantly downregulated in AGS NSCs compared to mouse NSCs under normal culture conditions. Data were presented as means ± S.D. *P* values calculated by unpaired two-tailed t-tests. ****$P < 0.0001$, n.s. indicates nonsignificant. $n = 3$ biological replicates. (C) Quantification of triglycerides level in AGS NSCs and mouse NSCs under normal culture conditions. Data were presented as means ± S.D. *P* values calculated by unpaired two-tailed t-tests. ***$P < 0.001$. $n = 3$ biological replicates. (D) Quantification of fatty acids in triglycerides in AGS NSCs and mouse NSCs showing that both saturated fatty acids and unsaturated fatty acids were significantly downregulated in AGS NSCs compared to mouse NSCs under normal culture conditions. Data were presented as means ± S.D. *P* values calculated by unpaired two-tailed t-tests. ****$P < 0.0001$, ***$P < 0.001$. $n = 3$ biological replicates. (E) Representative confocal fluorescence images displaying single mid-plane views of lipid droplets and plasma membrane staining in mouse and AGS NSCs (left). Quantification of lipid droplets per mid-plane image in mouse and AGS NSCs (right). Scale bars: 10 μm. Data were presented as means ± SEM. *P* values calculated by unpaired two-tailed t-tests. n.s. indicates nonsignificant ($n = 30$ cells per condition). (F) Quantification of metabolites in AGS NSCs and mouse NSCs showing that malonic acid was significantly upregulated in AGS NSCs compared to mouse NSCs under normal conditions. Data were presented as means ± SEM. *P* values calculated by unpaired two-tailed t-tests. **$P < 0.01$, n.s. indicates nonsignificant. $n = 3$ biological replicates. (G, H) Schematic illustrating functional FAS (fatty acid synthesis) in the normal state of mouse NSCs and FAS-impaired state in AGS NSCs. Source data are available online for this figure.

2018; Ma et al, 2013; None et al, 2023), its roles in linking lipids to mitochondria and organismal hypoxia resilience remain largely unexplored. Understanding these mechanisms may provide new insights into metabolic adaptations and potential therapeutic targets for hypoxia-related diseases.

We previously discovered that a naturally occurring variant of the mitochondrial ATP synthase subunit ATP5G1 in AGS confers marked protection against hypoxia-induced cell death (Singhal et al, 2020). However, the role of lipids in AGS resilience to hypoxia remains largely unknown. In this study, we performed a lipidomic analysis and discovered that both saturated and unsaturated triglycerides are drastically reduced in AGS NSCs compared to mouse NSCs. Similarly, we found in *C. elegans* that fatty acid desaturases, fatty acid synthases and lipid storage were downregulated by HIF-1 in response to hypoxia. Notably, pharmacological inhibition of both saturated and unsaturated triglyceride biosynthesis conferred protection against hypoxia-induced cell injury in AGS or organismal death in *C. elegans*. Furthermore, the inhibition of triglyceride biosynthesis mitigated aging-related decline and alleviated APOE4-induced pathology in *C. elegans*. These findings reveal a conserved and previously unrecognized role of triglycerides in AGS NSCs and *C. elegans*, suggesting that targeting the fatty acid-triglyceride biosynthetic pathway may offer broad protective effects against hypoxia, aging, and APOE4-associated neurodegenerative pathologies.

## Results

### Downregulation of triglyceride and upregulation of malonic acid in AGS NSCs

To explore the potential contribution of lipid composition to the remarkable hypoxia resilience in AGS NSCs, we performed lipidomic analysis of AGS NSCs versus mouse NSCs under normal culture conditions (Fig. 1A; Methods). Among >100 common lipids profiled using high-performance liquid chromatography (HPLC) and mass spectrometry (MS), we found that triglycerides were strongly and specifically downregulated in AGS NSCs compared to mouse NSCs (Fig. 1B,C), while other lipids, including cholesteryl esters (CE), ceramides, sphingomyelin (SM), and most phospholipids remained largely unchanged (Dataset EV1). Triglycerides are lipid esters composed of glycerol and three fatty acids, typically

stored in lipid droplets as energy reserves. Both saturated fatty acids, including 14:0, 16:0, 16:1, and 18:0, as well as unsaturated fatty acids, including 18:1 and 18:2, in triglycerides were significantly downregulated (Fig. 1D). We also confirmed triglyceride downregulation in AGS NSCs as measured by triglyceride hydrolysis determination (Fig. 1C), although the overall number of lipid droplets does not apparently differ compared to that in mouse NSCs (Figs. 1E and EV1A).

To obtain a broader view on potential metabolic changes in AGS in addition to lipids and fatty acids, we further performed gas chromatography–mass spectrometry (GC-MS) metabolomic analysis of major metabolites and metabolic intermediates from AGS and mouse NSCs under normal culture conditions. We found that the fatty acid metabolic intermediate malonic acid was markedly upregulated in AGS NSCs compared to mouse NSCs, while the vast majority of other metabolites profiled, including most amino acids and sugars, are unchanged (Fig. 1F, Dataset EV2). Malonic acid is a key intermediate in fatty acid biosynthesis, and its steady-state accumulation is consistent with reduced triglyceride abundance in AGS, indicating downregulation of the fatty acid-to-triglyceride biosynthesis pathway. Since both mouse and AGS NSCs are cultured under identical normoxic conditions, these findings support an intrinsic genetic program driving the downregulation of fatty acid-triglyceride biosynthesis alongside the upregulation of malonic acid (Fig. 1G,H), which may facilitate adaptation to hypoxia in AGS cells (see below).

### Triglyceride downregulation upon acute exposure to hypoxia in *C. elegans*

In the de novo lipid biosynthetic pathway, acetyl-CoA is converted to malonyl-CoA, which is then converted to medium- or long-chain fatty acids by fatty acid synthase (FASN). Fatty acids can be synthesized as saturated triglycerides or as polyunsaturated fatty acids (PUFAs) by fatty acid desaturase (FAD) (Fig. 2A). The lipidomic and metabolomic profiles in intrinsically hypoxia-adapted AGS NSCs indicate that these key lipid biosynthetic enzymes might be regulated by hypoxia. To monitor FASN and FAT regulation under acute hypoxic stress in a non-AGS in vivo system, we took advantage of *C. elegans* to explore FASN and FAD regulation in live tissues. We found that *C. elegans* FASN-1 endogenously tagged with GFP was markedly downregulated when exposed to hypoxia (0.1%) for 24 h post the larval L4 stage

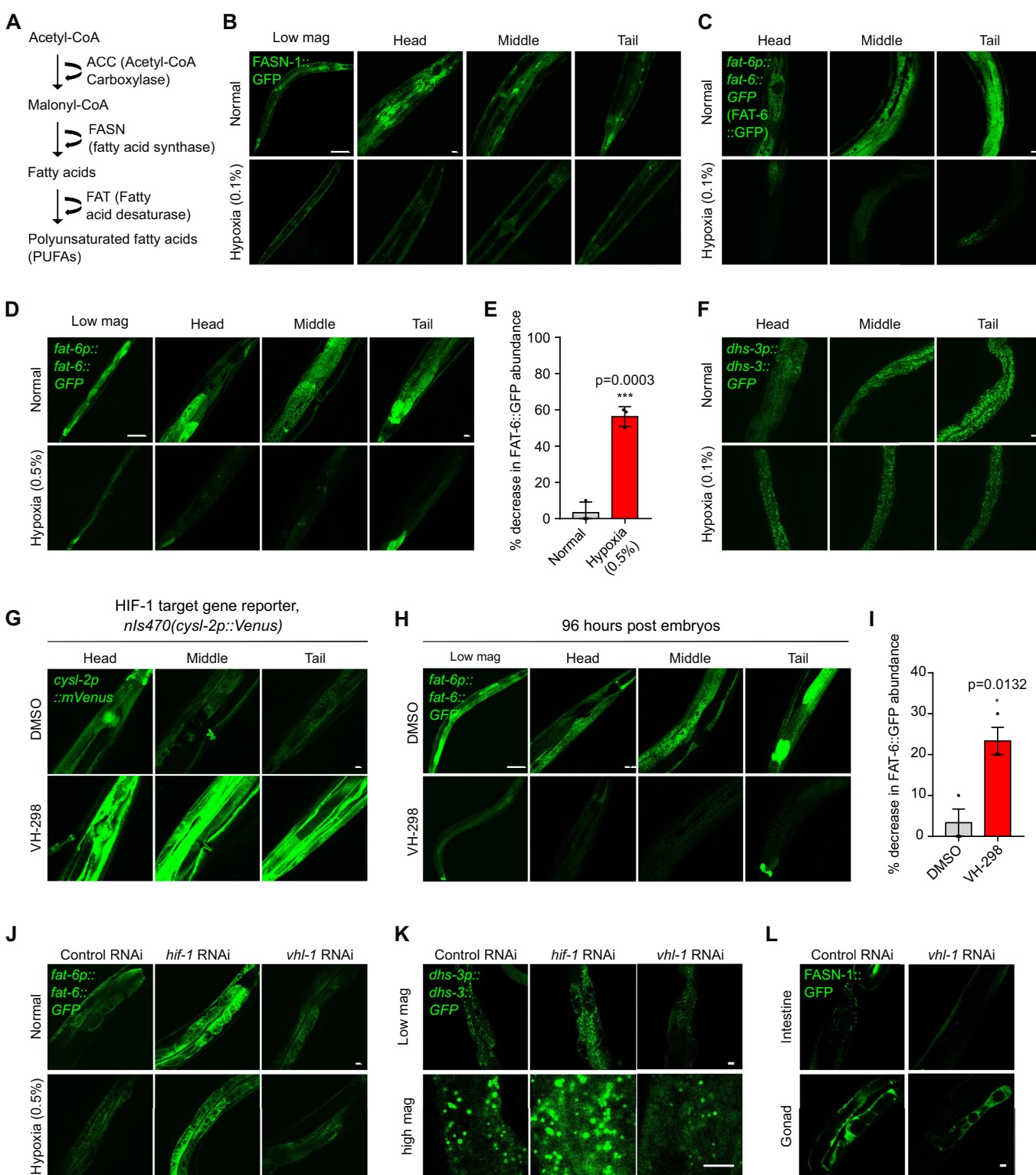

(Fig. 2B). We also generated a translational reporter *fat-6p::fat-6::GFP* by fusing GFP with the endogenous regulatory DNA sequences of *fat-6*, which encodes a major FAD in *C. elegans* (Watts and Ristow, 2017). Under normal conditions, FAT-6::GFP is basally expressed at high levels in various tissues, including the intestine

and hypodermis at L1-L4 stages, but is not detectable in neurons (Fig. EV2A,B). Exposure to hypoxia (0.1%) for 24 h post L4 stages resulted in marked downregulation of FAT-6::GFP abundance in the intestine (Fig. 2C). Similarly, exposure to hypoxia (0.5%) for 24 h post-L4 stages (Fig. 2D) caused more than 50% of worms with

◄ **Figure 2. Downregulation of unsaturated triglycerides and unsaturated lipid droplets by HIF-1 and hypoxia in *C. elegans*.**

(A) Schematic illustrating fatty acid synthesis and desaturation pathways. (B) Representative confocal fluorescence images showing the expression of FASN-1::GFP under normal or hypoxic (0.1%) conditions for 24 h post L4 stages. Scale bars: 100 μm (low mag) or 10 μm (high mag). (C) Representative confocal fluorescence images showing high-resolution Z-stack views of *fat-6p::fat-6::GFP* under normal or hypoxic (0.1%) conditions for 24 h post L4 stages. Scale bars: 10 μm. (D) Representative confocal fluorescence images showing low and high-resolution Z-stack views of *fat-6p::fat-6::GFP* under normal or hypoxic (0.5%) conditions for 24 h post L4 stages. Scale bars: 10 μm. (E) Quantification of the percentage of animals with *fat-6p::fat-6::GFP* fluorescence intensities downregulated under normal or hypoxic (0.5%) conditions for 24 h post L4 stages. Data were presented as means ± S.D. *P* values calculated by unpaired two-tailed t-tests. ***$P < 0.001$ (normal, $n = 30$ animals, hypoxic, $n = 37$ animals). (F) Representative confocal fluorescence images showing the downregulation of *dhs-3p::dhs-3::GFP*-labeled lipid droplet abundance and number under normal or hypoxic (0.1%) conditions for 24 h post L4 stages. Scale bars: 10 μm. (G) Representative confocal fluorescence images showing the expression of HIF-1-dependent transcriptional reporter, *nis470* (*cysl-2p::mVenus*) abundance under DMSO or VHL-1 inhibitor VH-298 (1 mg/ml) for 96 h post embryos stage. Scale bars: 10 μm. (H) Representative confocal fluorescence images showing low and high-resolution Z-stack views of *fat-6p::fat-6::GFP* under DMSO or VHL-1 inhibitor VH-298 (1 mg/ml) for 96 h post embryos stage. Scale bars: 10 μm. (I) Quantification of the percentage of animals with *fat-6p::fat-6::GFP* fluorescence intensities downregulated under DMSO or VHL-1 inhibitor VH-298 (1 mg/ml) for 96 h post embryos stage. Data were presented as means ± SEM. *P* values calculated by unpaired two-tailed t-tests. *$P < 0.05$ ($n = 30$ animals per condition). (J) Representative confocal fluorescence images showing *fat-6p::fat-6::GFP* animals fed with control RNAi or RNAi against *hif-1* or *vhl-1* under normal or hypoxic conditions. Scale bars: 10 μm. (K) Representative confocal fluorescence images showing *dhs-3p::dhs-3::GFP* animals fed with control RNAi or RNAi against *hif-1 or vhl-1* under normal conditions. Scale bars: 10 μm. (L) Representative confocal fluorescence images showing FASN-1::GFP animals fed with control RNAi or RNAi against *vhl-1* under normal conditions. Scale bars: 10 μm. Source data are available online for this figure.

downregulation at the 24-h time point (Fig. 2E). However, we observed that FAT-6::GFP was not significantly downregulated when exposed to relatively high oxygen levels (5%) for 24 h (Fig. EV2C,D), indicating that FAT-6::GFP downregulation is hypoxia-level dependent in *C. elegans*. Given that fatty acids can be converted to PUFAs by FAT-6, followed by biosynthesis as unsaturated triglycerides and transport to lipid droplets, we examined whether lipid droplets are affected by acute hypoxia treatment. Indeed, we found that the lipid droplet reporter (Zhang et al, 2012) strain *dhs-3p::dhs-3::GFP* showed marked downregulation in the number and size of lipid droplets abundance upon exposure to hypoxia (0.1%) for 24 h post L4 stages (Fig. 2F), and to a lesser extent with exposure to milder hypoxia (0.5%) (Fig. EV2F). These results reveal that acute hypoxia exposure leads to downregulation of FASN-1 and FAT-6 in *C. elegans* in a hypoxic level-dependent manner, accompanied by reduced lipid droplet number and size. This observation is consistent with overall reduced triglyceride levels observed in intrinsically hypoxia-adapted AGS NSCs (Fig. 1). The difference in lipid droplets likely reflects acute environmental hypoxic exposure in *C. elegans* versus intrinsic genetic adaptation in AGS NSCs, leading to differential lipid droplet regulation and compositions.

Next, we tested whether FAT and lipid droplet downregulation can be pharmacologically recapitulated by hypoxic activation of HIF-1 in *C. elegans*. As HIF-1 normally undergoes VHL-1-dependent proteolytic degradation, we employed an inhibitor (VH-298) of VHL-1 to protect against HIF-1 degradation (Frost et al, 2016). We confirmed that the HIF-1 target gene reporter *nIs470* abundance was upregulated by VH-298 when treated during early larval stages for 48 h (Fig. EV2G–I) or 96 h (Fig. 2G). We then examined FAT-6::GFP abundance changes in VH-298-treated worms and found that FAT-6::GFP was markedly downregulated in VH-298-treated worms compared to DMSO control (Fig. 2H,I). In addition, the abundance of FAT-6::GFP was upregulated by RNAi against *hif-1* and downregulated by RNAi against *vhl-1* (Figs. 2J and EV2E). Moreover, we found that the number of lipid droplets marked by DHS-3::GFP was increased by RNAi against *hif-1* and decreased by RNAi against *vhl-1* (Fig. 2K). The abundance of FASN-1::GFP in the gonad and associated somatic gonad was also downregulated by RNAi against *vhl-1* (Fig. 2L). Collectively, these data reveal hypoxia-induced and HIF-1-

dependent downregulation of lipid biosynthetic enzymes, which may lead to decreased unsaturated triglycerides and unsaturated lipid droplets in response to hypoxic stress.

## Inhibition of triglyceride biosynthesis and lipid droplet formation by HIF enhances hypoxia resilience in mouse neural cells and *C. elegans*

We also examined whether triglyceride and lipid droplet changes can be induced by ectopic expression of a non-degradable HIF-1 variant in non-AGS mammalian cells. We found that the number of lipid droplets was indeed markedly downregulated in the human embryonic kidney cell line HEK293T with non-degradable HIF-1 compared to the HEK293T cells (Fig. 3A). Since hypoxia and HIF-1 downregulate FASN-1 and FAT-6 in *C. elegans*, we next asked whether inhibition of fatty acid synthesis can bypass HIF activation in reducing lipid droplets and hypoxic injuries. Cerulenin, a natural antifungal antibiotic, effectively inhibits triglyceride biosynthesis by binding to fatty acid synthase, thereby disrupting the elongation of fatty acid chains essential for triglyceride formation (Price et al, 2001). We found that cerulenin treatment in mouse NSCs indeed led to lipid droplet reduction and markedly improved hypoxic survival in mouse NSCs (Figs. 3B–D and EV3A).

Besides direct FASN inhibition by cerulenin, we also employed the small-molecule drug TOFA (5-(Tetradecyloxy)-2-furoic acid) (McCune and Harris, 1979), which is a competitive inhibitor of acetyl-CoA carboxylase (ACC), a key enzyme involved in saturated and unsaturated fatty acid-triglyceride biosynthesis (Fig. EV3B). We found that TOFA decreased both the number and size of lipid droplets in mouse NSCs, as revealed by lipid droplet staining (Fig. 3B). In addition, TOFA alleviated acute hypoxia-induced membrane damage in mouse NSCs (Figs. 3E,F and EV3C,D), while it increased sensitivity to acute hypoxia in hypoxia-resilient AGS NSCs (Fig. 3G,H). These findings suggest that excessive triglycerides in mouse NSCs and insufficient triglycerides in AGS NSCs are both detrimental under hypoxic conditions, highlighting that an optimal level of triglycerides may be beneficial for cell survival against hypoxic stress.

We also explored the impact of triglyceride-related enzymes during hypoxic stress in vivo. The *C. elegans* genome encodes one ortholog of FASN and three paralogs of FAD, the complete lack of which is lethal

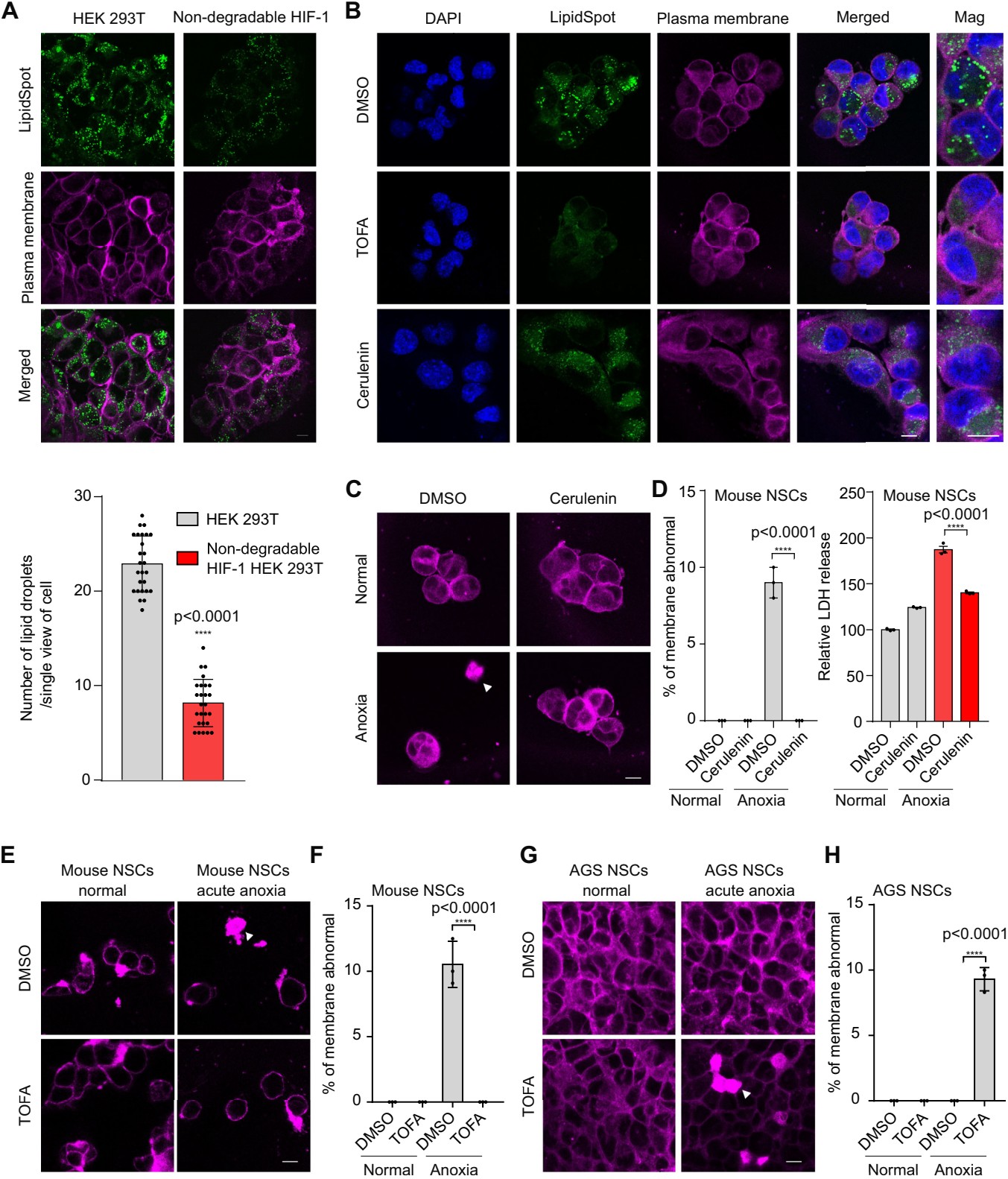

**Figure 3.   HIF activation or pharmacological inhibition of fatty acid biosynthesis reduces cellular lipid droplets and hypoxic injuries.**

(A) Representative confocal microscopy images and quantification of lipid droplets (green) and plasma membrane (magenta) in HEK293T cells and a non-degradation HIF-1 ectopic expression HEK293T cell line. Scale bar: 10 μm. Data were presented as means ± S.D. $P$ values calculated by unpaired two-tailed t-tests. ****$P < 0.0001$ ($n = 26$ cells per condition). (B) Representative confocal microscopic images of lipid droplets (green), plasma membrane (magenta), and DAPI (blue) in mouse NSCs treated with DMSO, or 20 nM TOFA, 20 nM Cerulenin, or alpha ketoglutarate for 16 h. Scale bar: 10 μm. (C, D) Representative confocal fluorescence images of membrane staining and quantification of the percentage of membrane abnormal (data were presented as means ± S.D.) or LDH release (data were presented as means ± SEM) under DMSO or Cerulenin, for 16 h in mouse NSCs, followed by normal or acute hypoxia for 60 min. Scale bar: 10 μm. $P$ values calculated by two-way ANOVA. ****$P < 0.0001$ ($n > 30$ cells per condition for membrane abnormal). (E, F) Representative confocal fluorescence images of membrane staining and quantification of the percentage of membrane damage under DMSO or the ACC inhibitor TOFA for 16 h in mouse NSCs, followed by normal or acute hypoxia for 60 min. Scale bar: 10 μm. Data were presented as means ± S.D. $P$ values calculated by two-way ANOVA. ****$P < 0.0001$ ($n > 30$ cells per condition). (G, H) Representative confocal fluorescence images of membrane staining and quantification of the percentage of membrane damage under DMSO or the ACC inhibitor TOFA for 16 h in AGS NSCs, followed by normal or acute hypoxia for 60 min. Scale bar: 10 μm. Data were presented as means ± S.D. $P$ values calculated by two-way ANOVA. ****$P < 0.0001$ ($n > 30$ cells per condition). Source data are available online for this figure.

(Watts and Ristow, 2017; Lee et al, 2010). We found that reduction-of-function alleles of *fasn-1(g14)* and loss-of-function double mutants of *fat-6(tm331); fat-5(tm420)* both decreased worm death under long-term hypoxic stress conditions (Fig. 4A). Next, we cultured WT *C. elegans* embryos on DMSO- or TOFA-supplemented NGM plates to L4 stages (Fig. EV4A), followed by long-term hypoxia (0.5%) for 72 h. We found that TOFA can protect against hypoxia-induced organismal death compared to DMSO (Fig. 4B). To explore the mechanism of triglyceride downregulation linked to hypoxia resilience, we examined the cellular consequences post administration of TOFA. We found that TOFA decreased the size of lipid droplet (Fig. 4C) and slightly decreased the *fat-6p::fat-6::GFP* fluorescence intensity under normal conditions (Fig. EV4B). Moreover, we found that short-term hypoxia (0.5% $O_2$ for 24 h) induced a shift in MAI-2::GFP-labeled mitochondrial morphology from fusion to fission (Fig. 4D). TOFA treatment decreased the degree of mitochondrial fission after hypoxia, compared to DMSO, particularly in energetic demanding body wall muscle cells (Fig. 4E,F). To further confirm this phenotype, we employed another mitochondrial marker mCherry::PDR-1, a protein that helps maintain mitochondrial homeostasis by tagging damaged mitochondria for destruction (Cooper and Van Raamsdonk, 2018; Vozdek et al, 2022). We found that short-term hypoxia (0.5% for 24 h) caused a shift in mCherry::PDR-1-labeled mitochondria morphology from fusion to fission as well (Fig. EV4C). Similar to MAI-2::GFP-labeled mitochondria, TOFA alleviated mitochondrial fission post hypoxia (0.5% for 24 h) compared to DMSO, based on mCherry::PDR-1-labeled mitochondria (Fig. 4G). Taken together, these data suggest that triglyceride downregulation protects against mitochondrial fission, contributing to the alleviation of death under hypoxia (Fig. 4H).

**Triglyceride biosynthetic inhibitor TOFA protects against APOE4-induced pathologies in *C. elegans***

We previously found that HIF-1 activation by the loss-of-function *vhl-1(ok161)* allele suppressed human lipoprotein APOE4-induced toxicity and neural pathologies in *C. elegans* (Jiang et al, 2025). Given that HIF-1-dependent FAT-6 downregulation leads to decreased triglycerides and lipid droplets in response to hypoxic stress, we next tested whether triglyceride reduction may also alleviate APOE4-induced pathologies. We employed TOFA to reduce both saturated and unsaturated triglyceride biosynthesis in strains expressing *hsp-16p::GFP*, a GFP reporter driven by a heat-shock promoter and abnormally upregulated by neuronal expression of APOE4 in *C. elegans* (Jiang et al, 2025). We found that

TOFA did not change the abundance or pattern of *hsp-16.2p::GFP* expression in muscles of wild-type animals (Fig. 5A). However, TOFA strongly attenuated the overall abundance of *hsp-16.2p::GFP* in the APOE4-expressing strain without changing *unc-54p::mCherry* in body wall muscles, while decreasing the percentage of animals with positive *hsp-16.2p::GFP* (Fig. 5B). We also found that TOFA partially normalized APOE4-induced Q40::YFP aggregation but does not apparently impact WT (Fig. 5C–E). Moreover, TOFA protected against APOE4-induced mortality in *C. elegans* (Fig. 5F). These findings prompted us to test further whether TOFA could also alleviate APOE4-induced neurodegeneration. We treated the *APOE4; PVD::GFP* strain with TOFA starting at the embryo stage and found that TOFA partially rescued APOE4-induced PVD neurodegeneration (Fig. 5G,H) without apparently altering APOE4-induced lysosomal defects (Fig. EV5A). Together, these data reveal that triglyceride biosynthetic inhibitor TOFA can normalize human lipoprotein APOE4-induced pathologies in *C. elegans*, recapitulating effects of HIF-1 activation.

**Triglyceride downregulation protects against aging in *C. elegans***

Next, we explored the physiological impact of triglyceride regulation beyond hypoxic stress in *C. elegans*. We found that FAT-6::GFP abundance was downregulated during starvation and aging at 25 °C, 20 °C or 15 °C in *C. elegans* (Figs. 6A,C and EV6A–C). Moreover, FASN-1::GFP was also downregulated during starvation (Fig. EV6D). We examined the lipid droplet reporter DHS-3::GFP during aging and found that DHS-3::GFP-labeled lipid droplet numbers were markedly downregulated during aging at 25 °C, 20 °C or 15 °C (Fig. 6B,D) and starvation (Fig. EV6E). We measured the lifespans of WT animals under DMSO control or TOFA-treated conditions starting at embryos under different common temperatures. We found that TOFA markedly extended *C. elegans* median lifespans (Table EV1) at 25 °C (Fig. 6E), 20 °C (Fig. 6F), and 15 °C (Fig. 6G). Together, these data indicate that triglyceride downregulation by TOFA also protects against aging trajectories in *C. elegans*.

# Discussion

Our study discovers triglyceride downregulation as a convergent mechanism conferring hypoxia resilience in AGS, neuroprotection

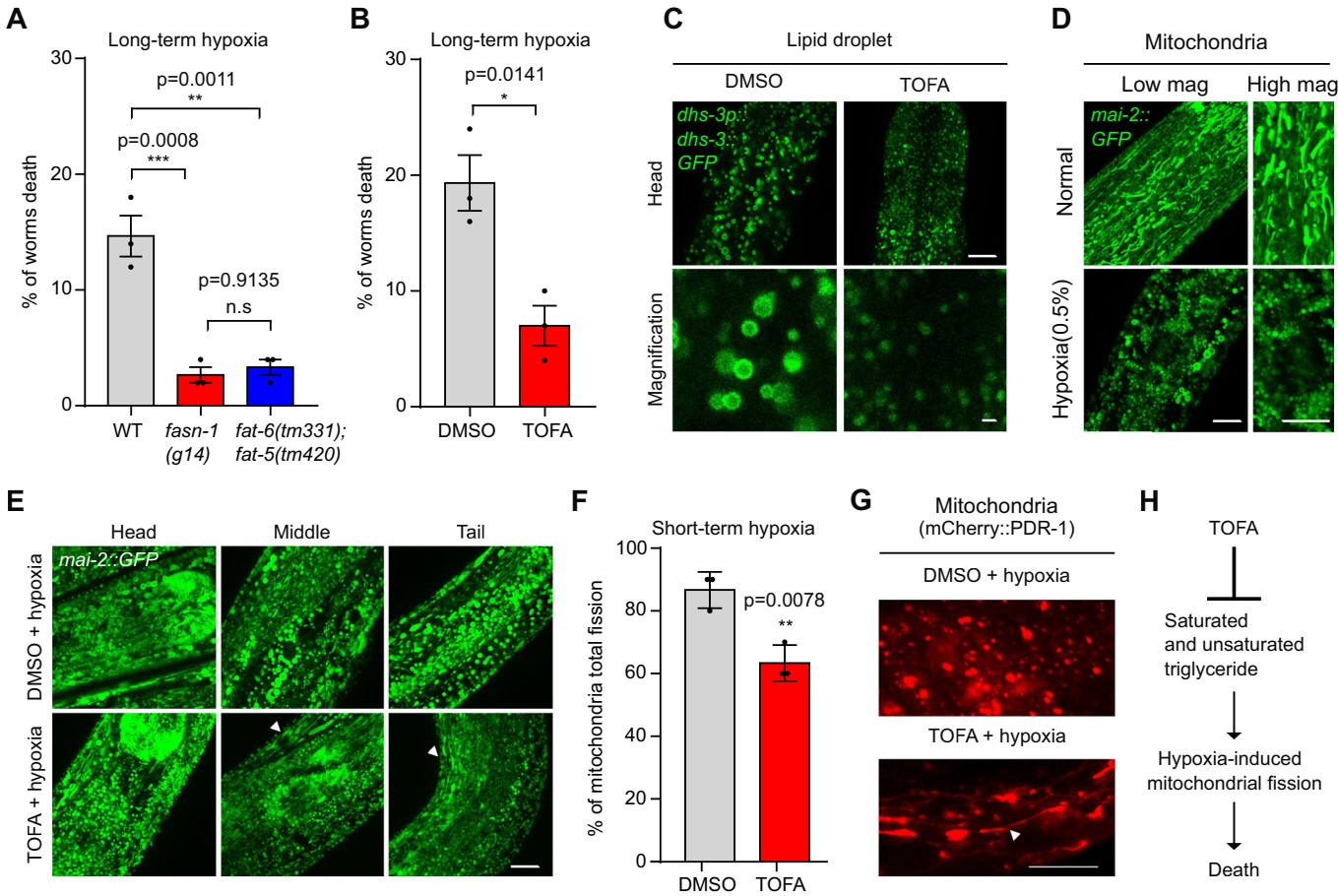

**Figure 4. Pharmacological inhibition of triglyceride biosynthesis protects against hypoxia in *C. elegans*.**

(A) Quantification of the percentage of worm death in WT or mutant alleles *fasn-1(g14)* or *fat-6(tm331); fat-5(tm420)* under long-term hypoxia conditions for 72 h. Data were presented as means ± SEM. *P* values calculated by one-way ANOVA. ***$P < 0.001$; **$P < 0.01$, n.s indicates nonsignificant ($n > 150$ worms per condition). (B) Quantification of the percentage of worm death under DMSO or the ACC inhibitor TOFA starting at embryos for 48 h, followed by long-term hypoxia for 72 h. Data were presented as means ± SEM. *P* values calculated by unpaired two-tailed t-tests. *$P < 0.05$ ($n > 150$ worms per condition). (C) Representative confocal fluorescence images showing high-resolution Z-stack views of *dhs-3p::dhs-3::GFP* under DMSO or the ACC inhibitor TOFA for 72 h (YA) post-embryo stage. Scale bars: 10 µm or 1 µm (magnification). (D) Representative confocal fluorescence images showing high-resolution Z-stack views of *mai-2::GFP*-labeled mitochondrial morphology under normal or short-term hypoxic (0.5%) conditions for 24 h post-L4 stages. Scale bars: 10 µm. (E) Representative confocal fluorescence images showing high-resolution Z-stack views of *mai-2::GFP*-labeled mitochondrial morphology, with effect more pronounced in body wall muscles (arrows) by exposure to the ACC inhibitor TOFA for 48 h starting from the embryo stage, followed by short-term hypoxia for 24 h. Scale bars: 10 µm. (F) Quantification of the percentage of mitochondria with "total" fission in *mai-2::GFP*-labeled mitochondria under DMSO or the ACC inhibitor TOFA for 48 h starting from the embryo stage, followed by short-term hypoxia for 24 h. Data were presented as means ± S.D. *P* values calculated by unpaired two-tailed t-tests. **$P < 0.01$ ($n > 25$ animals per condition). (G) Representative confocal fluorescence images showing high-resolution Z-stack views of mCherry-PDR-1-labeled mitochondria morphology under DMSO or the ACC inhibitor TOFA for 48 h starting from the embryo stage, followed by short-term hypoxia for 24 h. Scale bars: 10 µm. (H) Schematic model showing that the ACC inhibitor TOFA-induced downregulation of saturated and unsaturated triglyceride protects against mitochondrial fission, contributing to the alleviation of organismal death. Source data are available online for this figure.

against APOE4-associated pathologies and extended lifespan in *C. elegans* (Fig. 7A–C). By integrating lipidomic and metabolomic profiling of AGS cells with functional mechanistic studies in *C. elegans*, we establish a previously underappreciated mechanism of resilience to hypoxia and aging. The evolutionary conservation of this mechanism in AGS and *C. elegans* supports its biological importance and broad implication. Our findings suggest that targeting triglyceride biosynthetic pathways could yield potential therapeutic interventions for ischemic conditions, neurodegenerative diseases, and aging-related disorders.

Our mechanistic model posits that hypoxia triggers HIF-1-dependent suppression of triglyceride biosynthesis, which we

identify as a critical adaptive strategy for cellular survival. Suppression of triglyceride biosynthesis may represent an adaptive strategy to limit oxygen-intensive mitochondrial β-oxidation and ROS production. Triglycerides can serve as a fuel reserve for synapse function in neurons under normal conditions (Kumar et al, 2025) and provide a crucial energy source for other cells during periods of chronic energy deficit (Zadoorian et al, 2023). By downregulating triglyceride, cells under acute ischemic or hypoxic stresses may better preserve bioenergetic balance and anti-oxidative responses. Consistent with this, we find that reduced triglyceride levels mitigate hypoxia-induced mitochondrial fission, a stress response often linked to energetic failure. In AGS NSCs, we

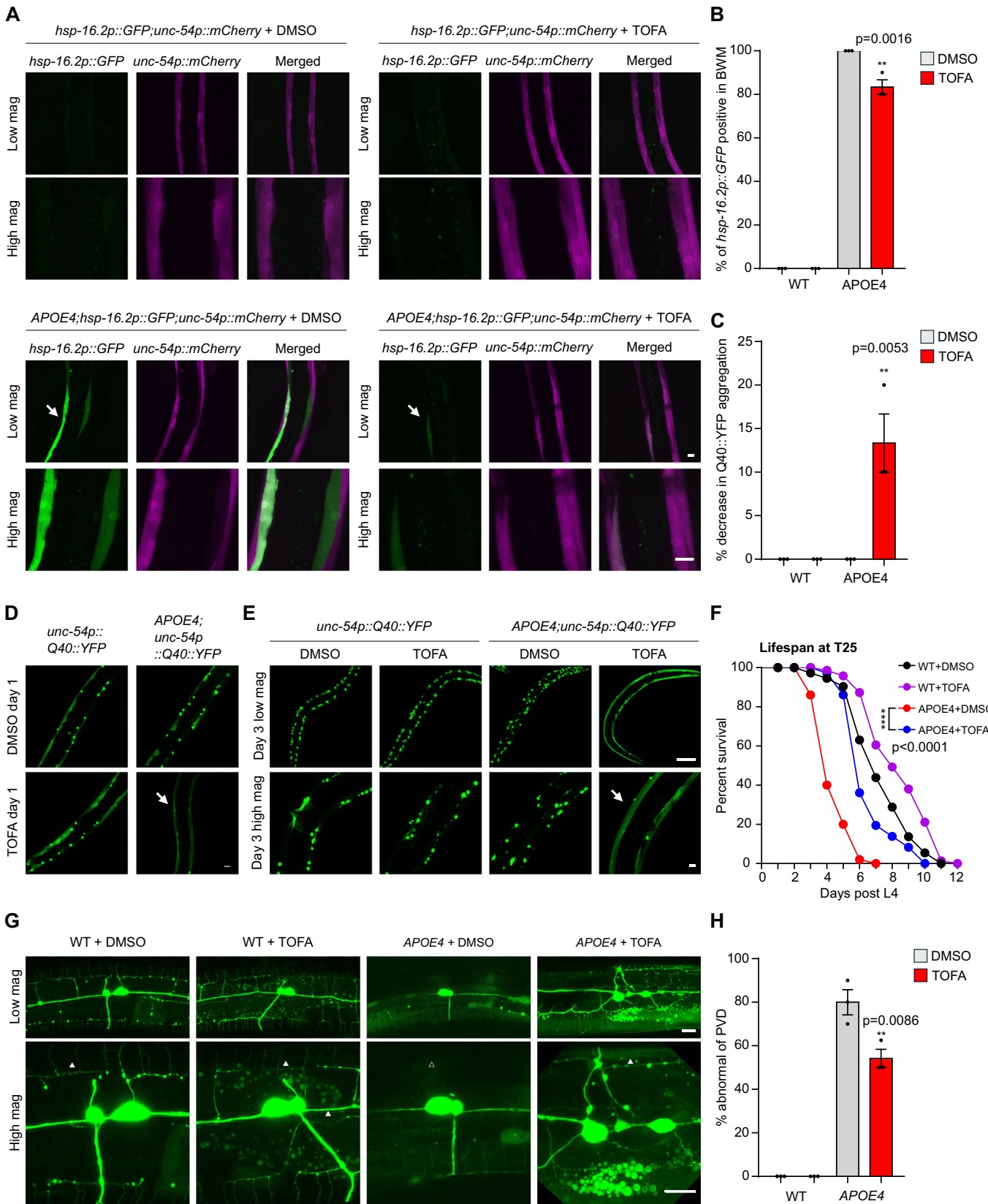

**Figure 5. Downregulation of saturated and unsaturated triglycerides by TOFA protects against APOE-induced pathologies in *C. elegans*.**

(A) Representative low and high magnification confocal microscopic images of *hsp-16.2p::GFP; unc-54p::mCherry* or *APOE4; hsp-16.2p::GFP; unc-54p::mCherry* (white arrows) at the YA stage, treated with DMSO or 1 mg/ml TOFA starting at the embryo stage. Scale bar: 10 μm. (B) Quantification of the percentage of *hsp-16.2p::GFP* positive expression in the BWM of WT or APOE4 at the YA stage, treated with DMSO or 1 mg/ml TOFA starting at the embryo stage. Data were presented as means ± SEM. *P* values calculated by two-way ANOVA. **$P < 0.01$ ($n > 30$ animals per condition). (C) Quantification of the percentage of decrease in Q40::YFP aggregation in the BWM of WT or APOE4 at the Day3 post L4 stage, treated with DMSO or 1 mg/ml TOFA starting at the embryo stage. Data were presented as means ± SEM. *P* values calculated by two-way ANOVA. **$P < 0.01$ ($n > 30$ animals per condition). (D) Representative high magnification confocal microscopic images of *unc-54p::Q40::YFP* or *APOE4; unc-54p::Q40::YFP* (white arrows) at the YA stage, treated with DMSO or 1 mg/ml TOFA starting at the embryo stage. Scale bar: 10 μm. (E) Representative low and high magnification confocal microscopic images of *unc-54p::Q40::YFP* or *APOE4; unc-54p::Q40::YFP* (white arrows) at the day 3 post L4 stage, treated with DMSO or 1 mg/ml TOFA starting at the embryo stage. Scale bar: 100 μm (low mag) or 10 μm (high mag). (F) Lifespan curves of WT or *APOE4* animals treated with DMSO or TOFA starting at the embryo stage, followed by survival counting starting at L4 at 25 °C. Data were presented as means without error bars. The lifespan assay was quantified using Kaplan–Meier lifespan analysis, and *P* values were calculated using the log-rank test. **$P < 0.01$ ($n > 40$ animals per condition). (G) Representative confocal microscopic images of PVD neurons (*wyIs592[ser-2prom-3p::myr-GFP]*) in pan-neuronal expression of APOE4(*vxIs824*) animals at the young adult stage on NGM treated with DMSO or TOFA starting at the embryo stage, showing that TOFA rescues APOE4-induced PVD abnormalities with an apparent loss of the third and fourth branches. Solid and open triangles to indicate the presence and absence of PVD neurites, respectively. Scale bar: 10 μm. (H) Quantification of the percentage of PVD neurons that are abnormal (with the third and fourth branches of PVD neurons missing or severed) in pan-neuronal APOE4 animals at the young adult stage on NGM treated with DMSO or TOFA starting at the embryo stage. Data were presented as means ± SEM. *P* values calculated by two-way ANOVA. **$P < 0.01$ ($n > 30$ animals per condition). Source data are available online for this figure.

observed reduced triglyceride levels alongside elevated malonic acid, indicating hypoxia-adapted reduction of fatty acid synthase activity, which likely reduces lipotoxicity and stabilizes cellular metabolism under low-oxygen conditions. Similarly, in *C. elegans*, we found that acute hypoxia induces HIF-1-mediated downregulation of lipid biosynthetic enzymes (FASN-1 and FAT-6). By pharmacologically inhibiting triglyceride biosynthesis, we confirmed that this reduction suppresses hypoxia-induced mitochondrial fission, promoting mitochondrial, cellular and organismal resilience. We previously discovered that cholesterol played an important role in APOE4-induced cellular damage suppressed by HIF-1 activation (Jiang et al, 2025). As major constituents of lipid droplets and lipoprotein particles, triglyceride and cholesterol may jointly contribute the cellular damage caused by APOE4 or other oxidative stress conditions. These findings collectively suggest that triglyceride suppression is not merely a downstream consequence of hypoxia, but rather an evolutionarily conserved, HIF-1-directed metabolic reprogramming that confers resilience and adaptive advantage. This strategy may reflect a broader principle wherein naturally hypoxia stress-resilient cells or organisms reallocate lipid flux away from storage and oxidation toward protective and reparative processes under hypoxic conditions. In this context, targeting lipid metabolic pathways, particularly triglyceride biosynthesis, may offer therapeutic leverage for enhancing mitochondrial integrity and stress resilience to treat ischemic, degenerative pathologic conditions in hypoxia-sensitive cells and organisms, including humans (Mutlu et al, 2021; Olzmann and Carvalho, 2019; Lopaschuk et al, 2010).

Despite the robustness of our findings, we acknowledge several limitations that merit further exploration. We relied on *C. elegans* and cultured AGS NSCs, which, while powerful for mechanistic dissection, may not fully recapitulate the complexity of in vivo mammalian systems. Our use of normoxic culture conditions for NSCs may overlook the dynamic physiological milieu of hibernation or hypoxia, potentially missing systemic or environmental influences on lipid metabolism. In other contexts, triglyceride accumulation supports synapse function in the brain (Kumar et al, 2025) and cancer cell proliferation in specific conditions of the tumor microenvironment (Röhrig and Schulze, 2016). Thus, inhibiting triglyceride biosynthesis may yield protective or detrimental outcomes depending on cell type, stressor, and metabolic demands. Although our findings suggest a triglyceride "sweet spot", the critical threshold remains to be quantitative defined and causally verified. We also do not yet know whether the selective upregulation of malonic acid plays a causal role beyond serving as a metabolic signature. Additionally, we employed TOFA as a pharmacological tool, but its unverified specificity in vivo raises the possibility of potential off-target effects, which could confound our mechanistic conclusions. We further note that sustained triglyceride downregulation may disrupt energy homeostasis or other lipid-dependent processes, a concern our study does not address. Future studies will address these gaps by exploring in vivo models and refining the specificity of lipid-targeted interventions.

## Methods

### Reagents and tools table

| Reagent/Resource | Reference or Source | Identifier or Catalog Number |
|---|---|---|
| **Bacterial strains** | | |
| Escherichia coli OP50 | Caenorhabditis Genetics Center (CGC) | OP50-NeoR |
| *E. coli* strain HT115 (DE3) | Source Bioscience | N/A |
| **Experimental models** | | |
| Cell lines | | |
| Human: 293 T | ATCC | CRL-3216 |
| Human: 293 T expression non-degradation HIF-1 | Ma lab | N/A |
| Mouse neural stem cell | Song Lab | N/A |
| AGS neural stem cell | Ma lab | N/A |
| **Recombinant DNA** | | |
| N/A | N/A | N/A |
| **Antibodies** | | |

| Reagent/Resource | Reference or Source | Identifier or Catalog Number |
|---|---|---|
| N/A | N/A | N/A |
| **Oligonucleotides and other sequence-based reagents** | | |
| N/A | N/A | N/A |
| **Chemicals, Enzymes and other reagents** | | |
| NeuroCult™ Basal Medium (Mouse & Rat) | STEMCELL | Cat# 05700 |
| NeuroCult™ Proliferation Supplement (Mouse & Rat) | STEMCELL | Cat# 05701 |
| Mouse EGF Recombinant Protein, PeproTech® | Thermo Fisher | Cat# 315-09 |
| Mouse FGF-basic (FGF-2/bFGF) Recombinant Protein, PeproTech® | Thermo Fisher | Cat# 450-33 |
| Heparin Solution | STEMCELL | Cat# 07980 |
| Cholesterol | Fisher Science | Cat# S25677 |
| Agarose | Sigma-Aldrich | Cat# A9539 |
| PFA (Paraformaldehyde) | Sigma-Aldrich | Cat# 16005 |
| Sodium azide | Sigma-Aldrich | Cat# 71290 |
| Ampicillin Sodium Salt | Fisher Scientific | Cat# BP1760-25 |
| IPTG | Millipore | Cat# 420322 |
| Triton X-100 | Sigma-Aldrich | Cat# T9284 |
| DMEM | Thermo Fisher | Cat# 11995065 |
| FBS (Fetal Bovine Serum) | Thermo Fisher | Cat# A5256701 |
| Opti-MEM | Thermo Fisher | Cat# 31985088 |
| PEI MAX | Polyscience | Cat# 24765 |
| LipidSpot™ Lipid Droplet Stains | Biotium | Cat# 70065 |
| CellMask™ Plasma Membrane Stains | Thermo Fisher | Cat# C10046 |
| TOFA | Sigma-Aldrich | Cat# 613450 |
| Cerulenin | Sigma-Aldrich | Cat# 219557 |
| VH-298 | MedChemExpress | Cat# HY-100947 |
| **Software** | | |
| GraphPad Prism | GraphPad Software | RRID: SCR_002798 |
| ImageJ | NIH | RRID: SCR_003070 |
| CaseViewer 2.4 | 3DHISTECH | RRID: SCR_017654 |
| Leica TCS SPE | Leica Microsystems | N/A |
| **Other** | | |
| N/A | N/A | N/A |

### C. elegans strains

*C. elegans* strains were grown on nematode growth media (NGM) plates seeded with *Escherichia coli* OP50 at 20 °C with laboratory standard procedures unless otherwise specified. The N2 Bristol strain was used as the reference wild type. Mutants and integrated transgenes were backcrossed at least 5 times. Genotypes of strains used (also in Table EV2): *dmaIs27[fat-6p::fat-6::GFP]; ldrIs1 [dhs-3p::dhs-3::GFP + unc-76(+)]; fasn-1(av138[fasn-1::gfp]), xmSi01[-mai-2p::mai-2::GFP]; dmaIs45[rpl-28p::mCherry::pdr-1], fasn-1(g14) I, fat-6(tm331) IV; fat-5(tm420) V, vxIs824 [rab-3p::ND18A-poE4::unc-54 3'UTR + myo-2p::mCherry::unc-54 3'UTR];wyIs592[-ser-2prom-3p::myr-GFP], vxIs824 [rab-3p::ND18ApoE4::unc-54 3'UTR + myo-2p::mCherry::unc-54 3'UTR];dmaIs8 [hsp-16.2p::GFP; unc-54p::mCherry];him-5(e1490), ceIs56 [unc-129p::ctns-1::mCherry + nlp-21p::Venus + ttx-3p::RFP], vxIs824 [rab-3p::ND18ApoE4::unc-54 3'UTR + myo-2p::mCherry::unc-54 3'UTR]; ceIs56 [unc-129p::ctns-1::mCherry + nlp-21p::Venus + ttx-3p::RFP], rmIs133 [unc-54p::Q40::YFP], vxIs824 [rab-3p::ND18ApoE4::unc-54 3'UTR + myo-2p::mCherry::unc-54 3'UTR]; rmIs133 [unc-54p::Q40::YFP].*

### Cell culture

Mouse NSCs (gift of Song lab, Baltimore, MD) and AGS NSCs (Neuronascent, Gaithersburg, MD, USA) have been previously described (Drew et al, 2016; Singhal et al, 2020). Both mouse NSCs and AGS NSCs grew under standard conditions at 37 °C and 5% $CO_2$ with NeuroCult basal media (STEMCELL, Vancouver, BC, CA) with 10% proliferation supplements (STEMCELL), FGF (100 ng/ml, PeproTech, Inc.), EGF (50 ng/ml, PeproTech, Inc., Rocky Hill, NJ, USA), and heparin (0.002%). Both mouse NSCs and AGS NSCs were cultured on 60 mm Petri dishes for maintenance. A full 60 mm Petri dish of mouse NSCs and AGS NSCs was collected as an individual repeat sample for lipidomic analysis.

### Lipidomic analysis of NSCs

Frozen mouse NSCs and AGS NSCs are stored until lipidomic extraction and analysis. All solvents for lipidomic extraction and analysis used were either HPLC or LC/MS grade and purchased from Sigma-Aldrich (St. Louis, MO, USA). Splash Lipidomix standards were purchased from Avanti (Alabaster, AL, USA). All lipid extractions were performed in 16 × 100 mm glass tubes with PTFE-lined caps (Fisher Scientific, Pittsburgh, PA, USA). Glass Pasteur pipettes and solvent-resistant plasticware pipette tips (Mettler-Toledo, Columbus, OH, USA) were used to minimize leaching of polymers and plasticizers. Samples were transferred to fresh glass tubes, and 1 mL of methanol, 1 mL of water, and 2 mL of methyl tert-butyl ether (mTBE) were added for liquid-liquid extraction. The mixture was vortexed and centrifuged at $2671 \times g$ for 5 min, resulting in two distinct liquid phases. The organic phase (upper phase) was transferred to a fresh tube with a Pasteur pipette and spiked with 20 μL of a 1:5 diluted Splash Lipidomix standard mixture. The samples were dried under $N_2$ and resuspended in 400 μL of hexane. Lipids were analyzed by LC-MS/MS using an SCIEX QTRAP 6500+ (SCIEX, Framingham, MA) equipped with a Shimadzu LC-30AD (Shimadzu, Columbia, MD) high-performance liquid chromatography (HPLC) system and a 150 × 2.1 mm, 5 μm Supelco Ascentis silica column (Supelco, Bellefonte, PA). Samples were injected at a flow rate of 0.3 mL/min at 2.5% solvent B (methyl tert-butyl ether) and 97.5% solvent A (hexane). Solvent B was increased to 5% over 3 min and then to 60% over 6 min. Solvent B

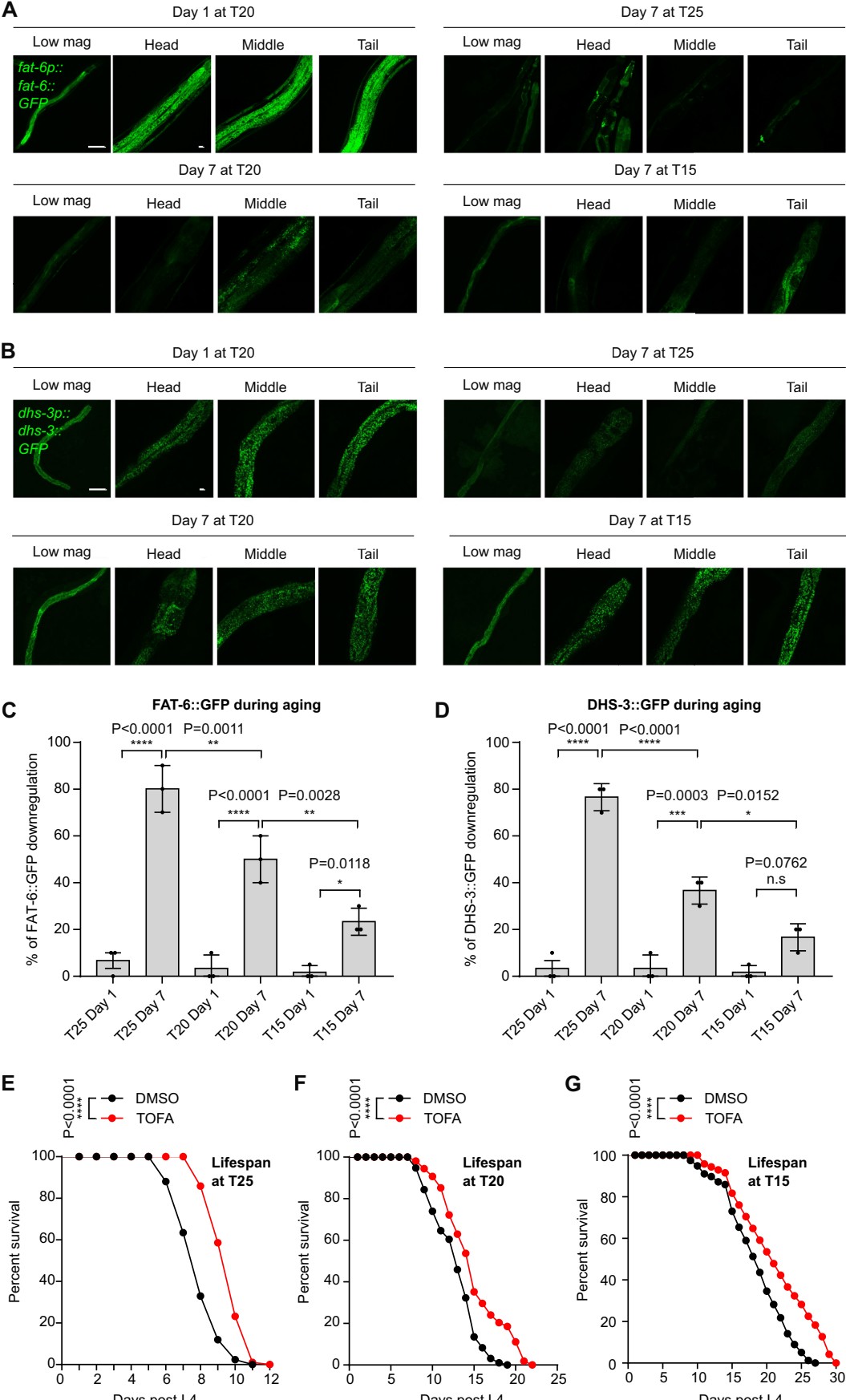

**Figure 6.   Downregulation of saturated and unsaturated triglyceride protect against aging in *C. elegans*.**

(A) Representative confocal fluorescence images showing high-resolution Z-stack views of *fat-6p::fat-6::GFP* under conditions of 7 days at T25, 7 days at T20, or 7 days at T15 post day 1 stage. Scale bars: 100 μm or 10 μm (magnification). (B) Representative confocal fluorescence images showing high-resolution Z-stack views of *dhs-3p::dhs-3::GFP* under conditions of 7 days at T25, 7 days at T20, or 7 days at T15 post day 1 stage. Scale bars: 100 μm or 10 μm (magnification). (C) Quantification of the percentage of animals with *fat-6p::fat-6::GFP* fluorescence intensities downregulated under conditions of 7 days at T25, 7 days at T20, or 7 days at T15 post day 1 stage. Data were presented as means ± S.D. $P$ values calculated by two-way ANOVA. *$P < 0.05$, **$P < 0.01$, ****$P < 0.0001$ ($n > 25$ animals per condition). (D) Quantification of the percentage of animals with *dhs-3p::dhs-3::GFP* fluorescence intensities downregulated under conditions of 7 days at T25, 7 days at T20, or 7 days at T15 post day 1 stage. Data were presented as means ± S.D. $P$ values calculated by two-way ANOVA. *$P < 0.05$, ***$P < 0.001$, ****$P < 0.0001$, n.s indicates nonsignificant ($n > 25$ animals per condition). (E) Lifespan curves of WT animals treated with DMSO or TOFA starting at the embryo stage, followed by survival counting starting at L4 at 25 °C. Data were presented as means without error bars. The lifespan assay was quantified using Kaplan–Meier lifespan analysis, and $P$ values were calculated using the log-rank test. ****$P < 0.0001$ ($n > 40$ animals per condition). (F) Lifespan curves of WT animals treated with DMSO or TOFA starting at the embryo stage, followed by survival counting starting at L4 at 20 °C. Data were presented as means without error bars. The lifespan assay was quantified using Kaplan–Meier lifespan analysis, and $P$ values were calculated using the log-rank test. ****$P < 0.0001$ ($n > 40$ animals per condition). (G) Lifespan curves of WT animals treated with DMSO or TOFA starting at the embryo stage, followed by survival counting starting at L4 at 15 °C. Data were presented as means without error bars. The lifespan assay was quantified using Kaplan–Meier lifespan analysis, and $P$ values were calculated using the log-rank test. ****$P < 0.0001$ ($n > 40$ animals per condition). Source data are available online for this figure.

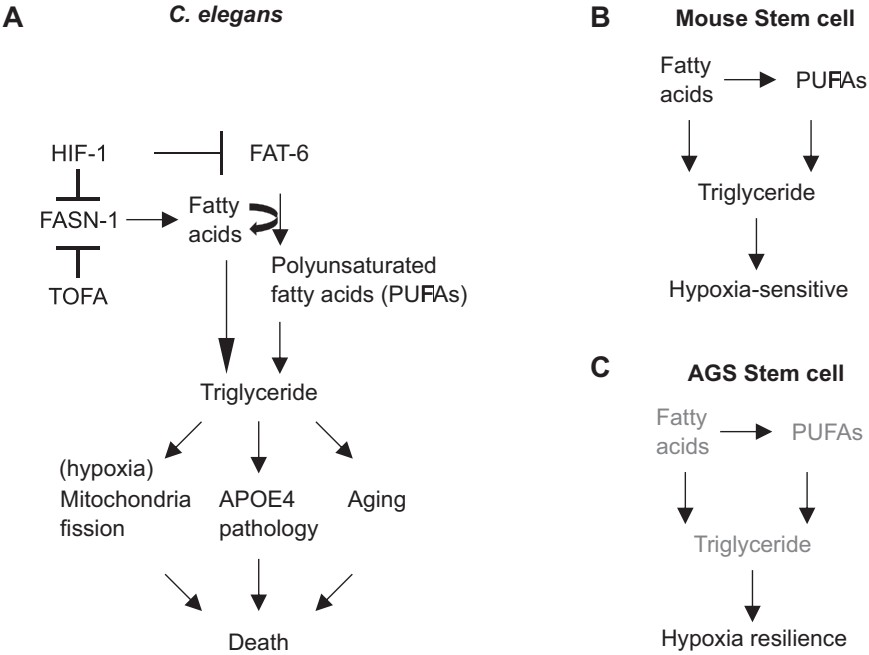

**Figure 7.   Model of triglyceride downregulation in protecting against hypoxic injury, APOE4-induced pathologies, and aging.**

(A) HIF-1 suppresses fatty acid biosynthesis and desaturation, leading to the downregulation of triglycerides to protect against hypoxia, APOE4 pathologies, and aging trajectories in *C. elegans*. Pharmacologically inhibiting TAG biosynthesis with TOFA can largely recapitulate these effects in HIF-1 activation in *C. elegans*. (B, C) Normal fatty acid biosynthesis and desaturation in mouse neural cells increase triglycerides, which can promote hypoxia sensitivity and injuries. In contrast, reduced triglyceride biosynthesis in AGS neural cells can contribute to hypoxia resilience. Other important players, including mitochondria and ROS, are omitted for clarity.

was decreased to 0% during 30 s, while solvent C (90:10 (v/v) isopropanol-water) was set at 20% and increased to 40% during the following 11 min. Solvent C was increased to 44% over 6 min and then to 60% over 50 s. The system was held at 60% solvent C for 1 min prior to re-equilibration at 2.5% solvent B for 5 min at a 1.2 mL/min flow rate. Solvent D [95:5 (v/v) acetonitrile-water with 10 mM ammonium acetate] was infused post-column at 0.03 mL/min. Column oven temperature was 25 °C. Data were acquired in positive and negative ionization modes using multiple reaction monitoring (MRM). The LC-MS/MS data were analyzed using MultiQuant software (SCIEX). The identified lipid species were normalized to their corresponding internal standard.

## Metabolomics analysis of NSCs

Metabolomics analysis was performed at the Metabolomics Core Facility at the University of Utah. Frozen mouse NSCs and AGS NSCs are stored until metabolomic extraction and analysis. Each sample was added cold 90% methanol (MeOH) solution to give a final concentration of 80% MeOH to each cell pellet. Samples were then incubated at −20 °C for 1 h. After incubation the samples were centrifuged at 20,000 × $g$ for 10 min at 4 °C. The supernatant was then transferred from each sample tube into a labeled, fresh micro centrifuge tube. Pooled quality control samples were made by removing a fraction of collected supernatants from each sample and

process blanks were made using only extraction solvent and no cell culture. The samples were then dried en vacuo.

All GC-MS analysis was performed with an Agilent 5977b GC-MS MSD-HES fit with an Agilent 7693 A automatic liquid sampler. Dried samples were suspended in 40 µL of a 40 mg/mL O-methoxyamine hydrochloride (MOX) (MP Bio #155405) in dry pyridine (EMD Millipore #PX2012-7) and incubated for 1 h at 37 °C in a sand bath. 25 µL of this solution was added to auto sampler vials. 60 µL of N-methyl-N-trimethylsilyltrifluoracetamide (MSTFA with 1% TMCS, Thermo #TS48913) was added automatically via the auto sampler and incubated for 30 min at 37 °C. After incubation, samples were vortexed and 1 µL of the prepared sample was injected into the gas chromatograph inlet in the split mode with the inlet temperature held at 250 °C. A 5:1 split ratio was used for analysis for the majority of metabolites. Any metabolites that saturated the instrument at the 5:1 split were analyzed at a 50:1 split ratio. The gas chromatograph had an initial temperature of 60 °C for 1 min followed by a 10 °C/min ramp to 325 °C and a hold time of 10 min. A 30-meter Agilent Zorbax DB-5MS with 10 m Duraguard capillary column was employed for chromatographic separation. Helium was used as the carrier gas at a rate of 1 mL/min. Below is a description of the two-step derivatization process used to convert non-volatile metabolites to a volatile form amenable to GC-MS.

Data was collected using MassHunter software (Agilent). Metabolites were identified and their peak area was recorded using MassHunter Quant. This data was transferred to an Excel spread sheet (Microsoft, Redmond, WA). Metabolite identity was established using a combination of an in-house metabolite library developed using pure purchased standards, the NIST library and the Fiehn library. There are a few reasons a specific metabolite may not be observable through GC-MS. The metabolite may not be amenable to GC-MS due to its size, or a quaternary amine such as carnitine, or simply because it does not ionize well. Metabolites that do not ionize well include oxaloacetate, histidine and arginine. Cysteine can be observed depending on cellular conditions. It often forms disulfide bonds with proteins and is generally at a low concentration. Metabolites may not be quantifiable if they are only present in very low concentrations.

## Transgenic arrays and strains

Transgenic animals that carry non-integrated, extra-chromosomal arrays were generated by co-injecting an injection marker with one to multiple DNA construct at 5–50 ng/µl. Animals that carry integrated transgenic arrays were generated from the animals by UV irradiation with 800 joules (UV Stratalinker2400, Stratagen), followed by outcrossing against N2 at least five times.

## Lipid droplet staining and triglyceride detection

Mouse NSCs and AGS NSCs grew on a coverslip. The cells were then incubated with a staining solution of LipidSpot™ 488 (Catalog no. 70065, Biotium) in medium (1:1000) at incubator for 10 min. The cells were imaged after three washes with PBS, followed by fixation with 4% PFA for 12 min at room temperature. Triglyceride levels were measured according to the manufacturer's instructions. Briefly, 150 µL of diluted Enzyme Mixture solution was added to

10 µL of cell samples taken from the cell culture wells. Absorbance was measured using a spectrophotometric plate reader.

## Cell membrane staining

Mouse NSCs and AGS NSCs were grown on a coverslip, and the cells were then incubated with a staining solution of CellMask™ Plasma Membrane Stains (C10046, Invitrogen™) in medium (1:1000) at incubator for 10 min, followed by fixation with 4% PFA for 12 min at room temperature. The cells were imaged after three washes with PBS.

## LDH release assay

Cell injury was also quantified by measuring lactate dehydrogenase (LDH) release from damaged cells according to the manufacturer's instructions. In brief, 50 µl of LDH reaction reagent was added to 50 µl conditional medium taken from the cell culture wells. The absorbance was measured at 490 nm by a spectrophotometer plate reader.

## Confocal imaging

Epifluorescence confocal microscopes (Leica TCS SPE) were used to capture fluorescence images (with a 10×, 20×, and 63× objective lens). Animals of different genotypes, stages, hypoxia treatments, and drug treatment were randomly selected and treated with a 10 mM sodium azide solution (71290-100MG, Sigma-Aldrich) in M9 buffer, aligned on an 2% agarose pad on slides for imaging. The same settings were maintained for the images of all samples.

## C. elegans hypoxia stress assay

Animals were cultured under non-starved conditions for at least 2 generations before hypoxia stress assays. The strain fasn-1(g14) was maintained at a low temperature of 15 °C. For short-term hypoxia expression patterns, synchronized L4 stage animals (n ≥ 50) were picked and transferred to new normal NGM plates seeded with OP50, then placed in a hypoxia incubator for 24 h set at different levels (0.1%, 0.5%, or 5%). For long-term hypoxia survival assays, animals were scored for survival post-exposure to hypoxia (0.5%) for 72 h. Animals failing to respond to repeated touches of a platinum wire were scored as dead.

## VH-298 treatment

Synchronized embryos (n ≥ 50) of different strains (cysl-2p::Venus or fat-6p::fat-6::GFP) were transferred to normal NGM plates seeded with OP50, supplemented with DMSO or 1 mg/ml VH-298, and incubated for 48–96 h at 20 °C for analysis.

## TOFA treatment

For cell culture, mouse NSCs and AGS NSCs were supplemented with DMSO or 20 nM TOFA for 16 h, followed by exposure to acute severe hypoxia (0.5%) for 60 min. For C. elegans expression analysis, synchronized embryos (n ≥ 50) of different strains (fat-6p::fat-6::GFP, mai-2::GFP, rpl-28p::mCherry::pdr-1, or APOE4;PVD) were transferred to normal NGM plates seeded with

OP50, supplemented with DMSO or 1 mg/ml TOFA, and incubated for 48–96 h at a normal temperature of 20 °C for analysis. For long-term hypoxia survival, synchronized embryos ($n \geq 50$) of N2 animals were transferred to normal NGM plates seeded with OP50, supplemented with DMSO or 1 mg/ml TOFA, followed by exposure to hypoxia (0.5%) for 72 h.

### RNA interference (RNAi)

RNAi were performed by feeding animals with *E. coli* strain HT115 (DE3) expressing double-strand RNA (dsRNA) targeting endogenous genes. Briefly, dsRNA-expressing bacteria were replicated from the Ahringer library to LB plates containing 100 μg/ml ampicillin (BP1760-25, Fisher Scientific) at 37 °C for 16 h. Single clone was picked to LB medium containing 100 μg/ml ampicillin at 37 °C for 16 h and positive clones (verified by bacteria PCR with pL4440 forward and pL4440 reverse primers) were spread onto NGM plates containing 100 μg/ml ampicillin and 1 mM isopropyl 1-thio-β-Dgalactopyranoside (IPTG, 420322, MilloporeMillipore) for 24 h. Developmentally synchronized embryos from bleaching of gravid adult hermaphrodites were plated on RNAi plates and grown at 20 °C followed by imaging.

### Lifespan analysis

For lifespan assays, animals were cultured under non-starved conditions for at least 2 generations before lifespan assays. For normal NGM lifespan assay, stage-synchronized L4 stages animals ($n \geq 50$) were picked to new NGM plates seeded with OP50 containing 50 μM 5-fluoro-2′-deoxyuridine (FUDR) to prevent embryo growth and bloating at 25 °C, 20 °C, or 15 °C. Animals were scored for survival per 24 h. Animals failing to respond to repeated touch of a platinum wire were scored as dead.

### Statistics

Data were analyzed using GraphPad Prism 9.2.0 Software (Graphpad, San Diego, CA) and presented as means ± S.D. unless otherwise specified, with *P* values calculated by unpaired two-tailed t-tests (comparisons between two groups), one-way ANOVA (comparisons across more than two groups) and two-way ANOVA (interaction between genotype and treatment), with post-hoc Tukey HSD and Bonferroni's corrections. The lifespan assay was quantified using Kaplan–Meier lifespan analysis, and *P* values were calculated using the log-rank test.

## Data availability

This study includes no data deposited in external repositories.

The source data of this paper are collected in the following database record: biostudies:S-SCDT-10_1038-S44319-025-00664-6.

## Peer review information

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

## Acknowledgements

Some strains were provided by the Caenorhabditis Genetics Center (CGC), which is funded by the NIH Office of Research Infrastructure Programs (P40 OD010440), and by Drs. Kang Shen and Navarro González. We also thank Wormbase.org (NIH grant #U24 HG002223 to P. Sternberg), Wormatlas.org (NIH grant #OD010943 to D.H. Hall), AgingAtlas and CenGen for the resources. Mass spectrometry equipment was obtained through NCRR Shared Instrumentation Grant 1S10OD016232-01, 1S10OD018210-01A1 and 1S10OD021505-01. The work was supported by NIH grants (R35GM139618 to DKM), AHA (24TPA1288391 to DKM), UCSF PBBR New Frontier Research Award (DKM), UCSF CIRM Scholars Training Program EDUC4-12812 (WIJ).

## Author contributions

**Wei I Jiang**: Conceptualization; Data curation; Formal analysis; Validation; Investigation; Visualization; Methodology; Writing—original draft; Writing—review and editing. **Goncalo Dias do Vale**: Data curation; Investigation; Methodology. **Quentinn Pearce**: Data curation; Formal analysis; Investigation; Methodology. **Kaitlyn Kong**: Data curation; Investigation. **Wenbing Zhou**: Data curation. **Jeffrey G McDonald**: Data curation; Funding acquisition; Investigation; Methodology. **James E Cox**: Supervision; Methodology. **Neel S Singhal**: Supervision; Methodology. **Dengke K Ma**: Conceptualization; Resources; Data curation; Formal analysis; Supervision; Funding acquisition; Investigation; Methodology; Writing—original draft; Project administration; Writing—review and editing.

Source data underlying figure panels in this paper may have individual authorship assigned. Where available, figure panel/source data authorship is listed in the following database record: biostudies:S-SCDT-10_1038-S44319-025-00664-6.

## Disclosure and competing interests statement

The authors declare no competing interests.

# Expanded View Figures

**A**

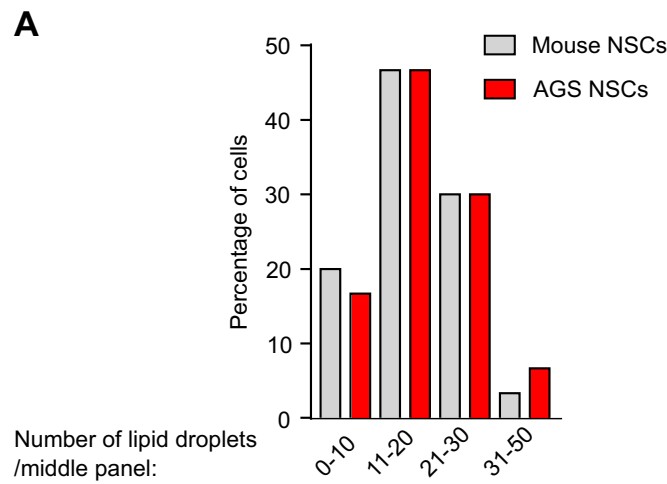

Number of lipid droplets
/middle panel:

**Figure EV1. Characterization of lipid droplets in AGS NSCs and mouse NSCs under normal conditions.**

(A) Quantification of the percentage of cells with varying numbers of lipid droplets per mid-plane image in mouse and AGS NSCs ($n > 25$ cells per condition).

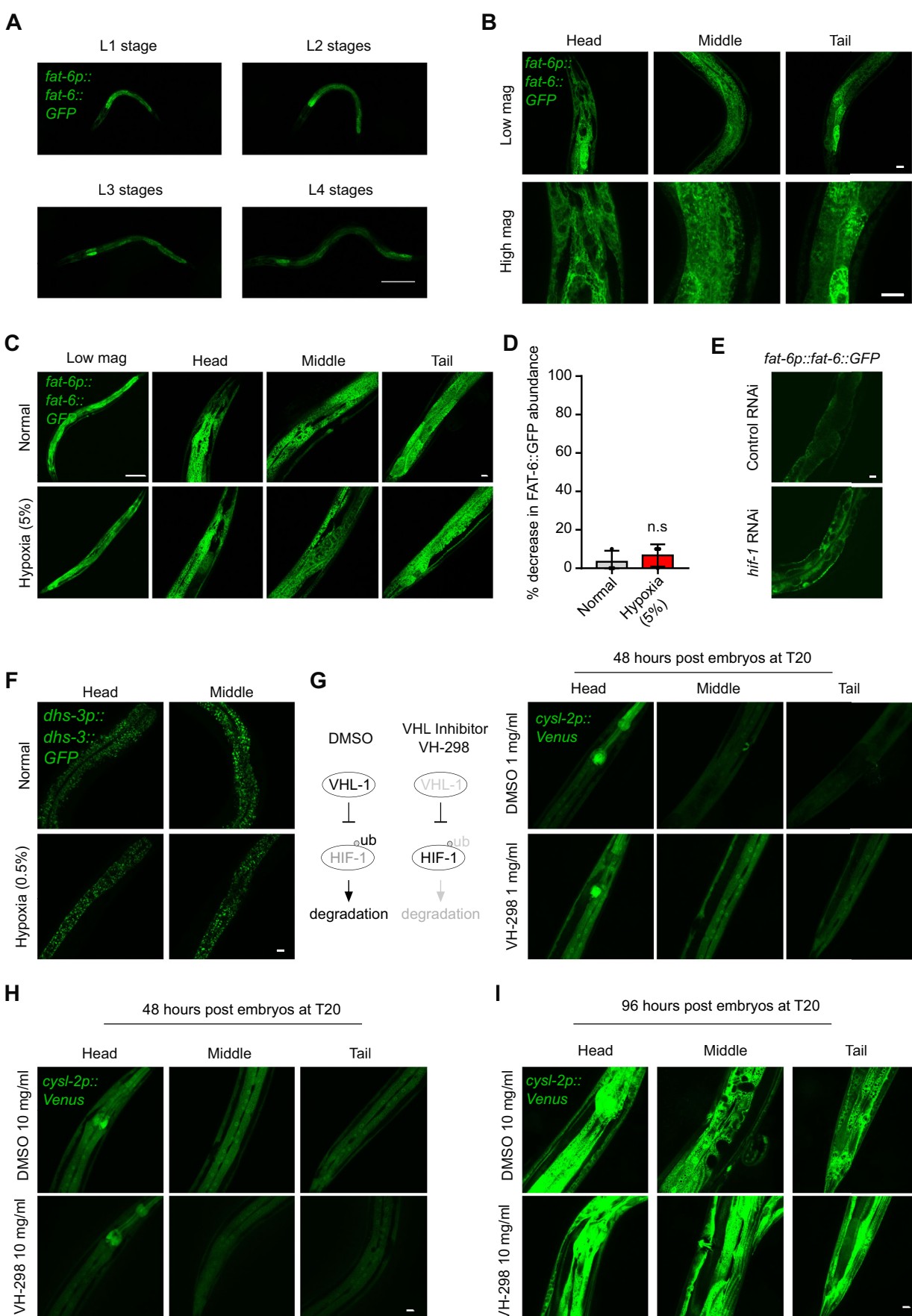

**Figure EV2.   Characterization of expression patterns of FAT-6 and lipid droplets with various conditions in *C. elegans*.**

(A) Representative confocal fluorescence images showing the expression of an integrated transgene *fat-6p::fat-6::GFP* under normal conditions at L1-L4 stages. Scale bars: 100 μm. (B) Representative Low and high magnification confocal fluorescence images showing the expression of *fat-6p::fat-6::GFP* under normal conditions L4 stages. Scale bars: 10 μm. (C) Representative confocal fluorescence images showing low and high-resolution Z-stack views of *fat-6p::fat-6::GFP* under normal or hypoxic (5%) conditions for 24 h post L4 stages. Scale bars: 10 μm. (D) Quantification of the percentage of animals with *fat-6p::fat-6::GFP* fluorescence intensities downregulated under normal or hypoxic (5%) conditions for 24 h post L4 stages. Data were presented as means ± S.D. *P* values calculated by unpaired two-tailed t-tests. n.s indicates nonsignificant ($n > 25$ animals per condition). (E) Representative confocal fluorescence images showing *fat-6p::fat-6::GFP* fed with control RNAi or RNAi against *hif-1*. Scale bars: 10 μm. (F) Representative confocal fluorescence images showing the downregulation of *dhs-3p::dhs-3::GFP*-labeled lipid droplet number and size under normal or hypoxic (0.5%) conditions for 24 h post-L4 stages. Scale bars: 10 μm. (G) Schematic diagram of HIF-1, VHL-1, and the drug VH-298. Representative confocal fluorescence images showing the upregulation of HIF-1 target reporter *cysl-2p::Venus* treat with DMSO or VHL-1 inhibitor VH-298 (1 mg/ml) for 48 h starting at embryos. Scale bars: 10 μm. (H) Representative confocal fluorescence images showing the upregulation of HIF-1 target reporter *cysl-2p::Venus* treat with DMSO or VHL-1 inhibitor VH-298 (10 mg/ml) for 48 h starting at embryos. Scale bars: 10 μm. (I) Representative confocal fluorescence images showing the upregulation of HIF-1 target reporter *cysl-2p::Venus* treat with DMSO or VHL-1 inhibitor VH-298 (10 mg/ml) for 96 h starting at embryos. Scale bars: 10 μm.

**A**

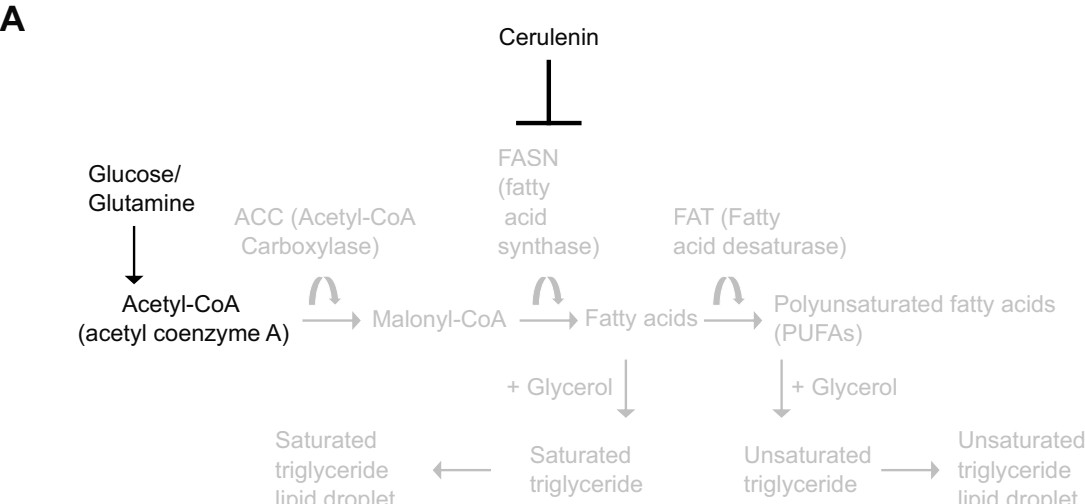

**B**

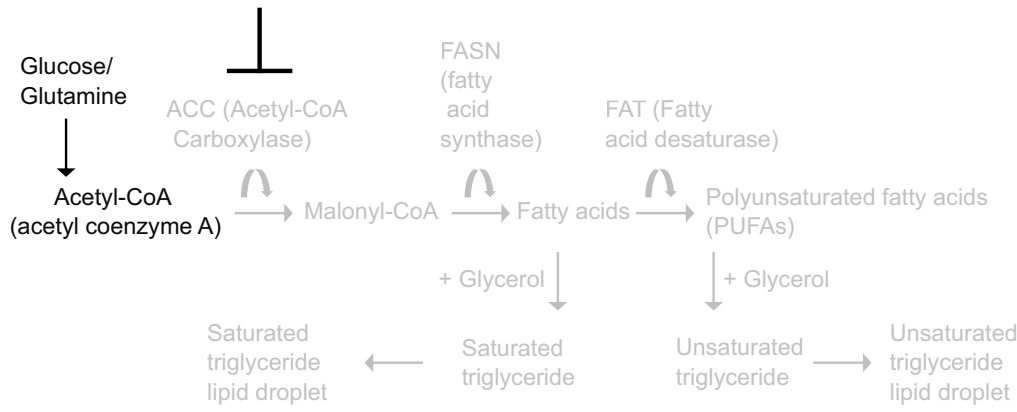

**C**

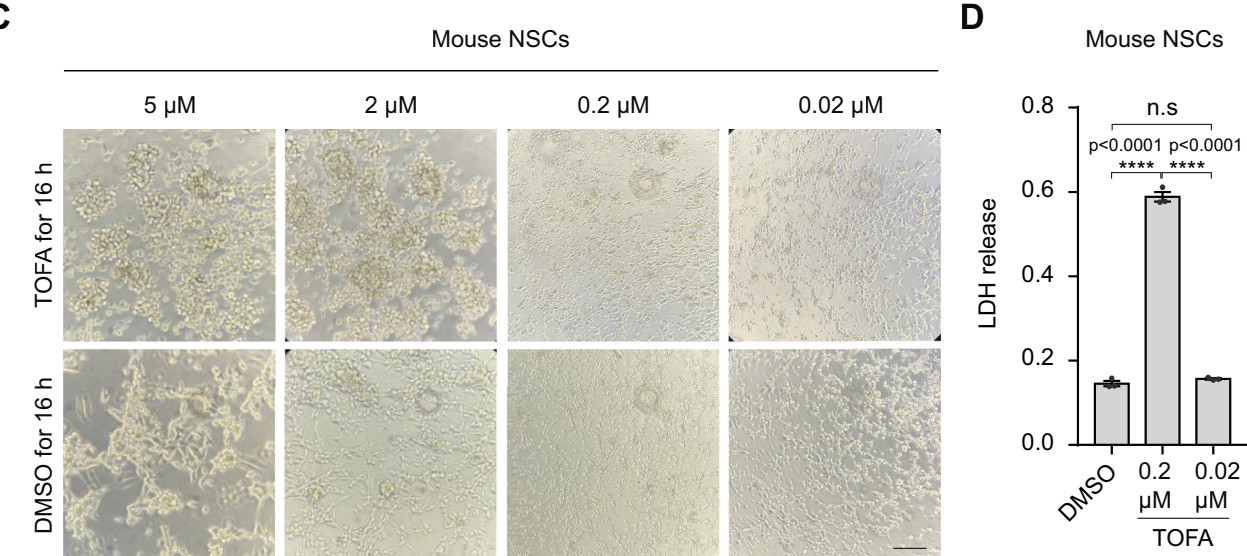

◀ **Figure EV3. Characterization of TOFA in mouse neural stem cell.**

(A, B) Schematic of FASN inhibitor cerulenin and acetyl-CoA carboxylase (ACC) inhibitor TOFA (5-(Tetradecyloxy)-2-furoic acid), which blocks both saturated and unsaturated triglyceride biosynthesis. (C, D) Representative images and LDH release of mouse NSCs treated with different concentrations of TOFA or DMSO for 16 h. Data were presented as means ± SEM. $P$ values calculated by one-way ANOVA. ****$P < 0.0001$, n.s indicates nonsignificant. $n = 3$ biological replicates. Scale bar: 100 μm.

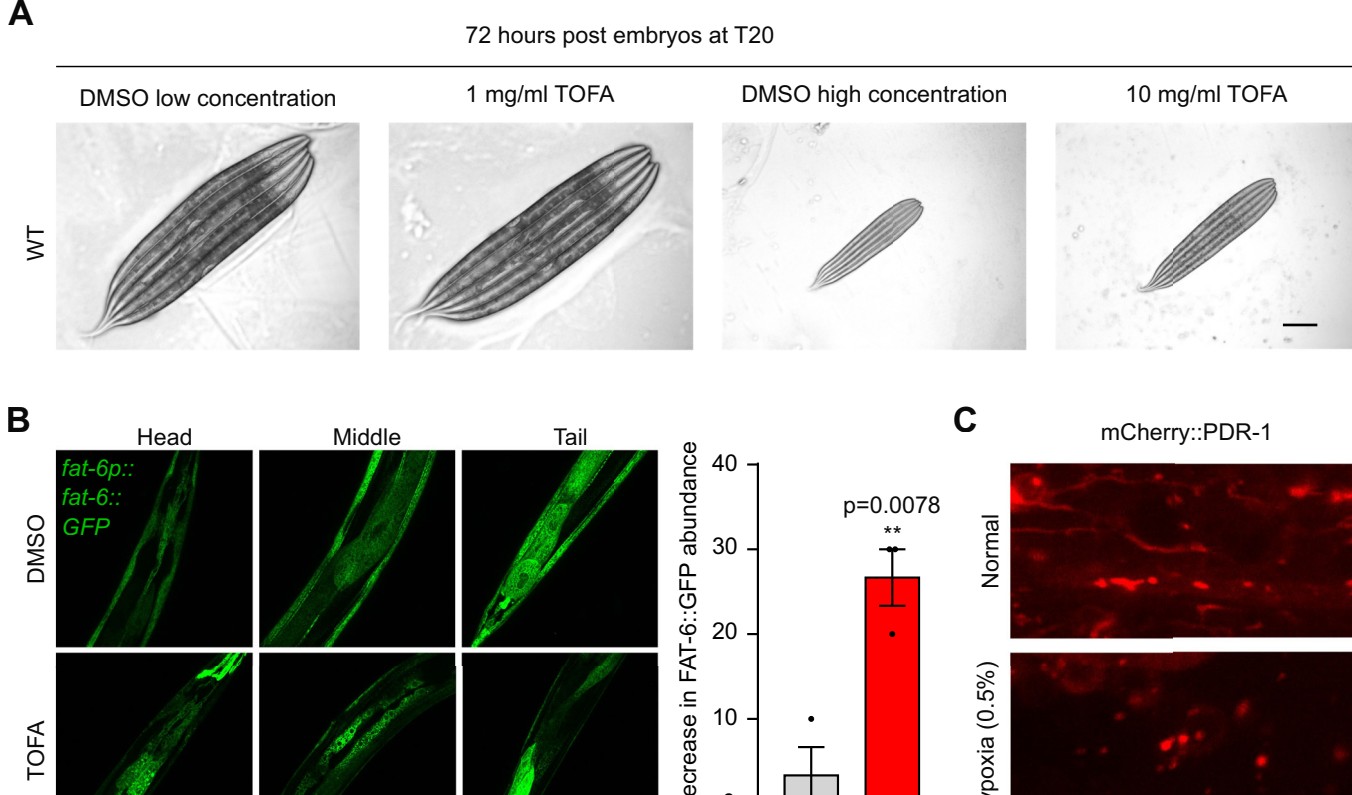

**A**

72 hours post embryos at T20

**B**

**C**

mCherry::PDR-1

Figure EV4.    Characterization of TOFA and expression patterns of mCherry::PDR-1 upon exposure to short-term hypoxia in *C. elegans*.

(**A**) Representative images of body size in wild-type N2 animals treated with varying concentrations of DMSO or TOFA, starting at the embryo stage at 20 °C. Scale bar: 100 μm. (**B**) Representative confocal fluorescence images and quantification of the percentage of animals with *fat-6p::fat-6::GFP* fluorescence intensities downregulated treated with DMSO or 1 mg/ml TOFA starting at the embryo stage. Scale bar: 10 μm. Data were presented as means ± S.D. *P* values calculated by unpaired two-tailed t-tests. **$P < 0.01$ ($n = 30$ animals per condition). (**C**) Representative confocal fluorescence images showing high-resolution Z-stack tail area views of mCherry-PDR-1-labeled mitochondria morphology under normal or hypoxia (0.5%) for 24 h post-L4 stages. Scale bars: 10 μm.

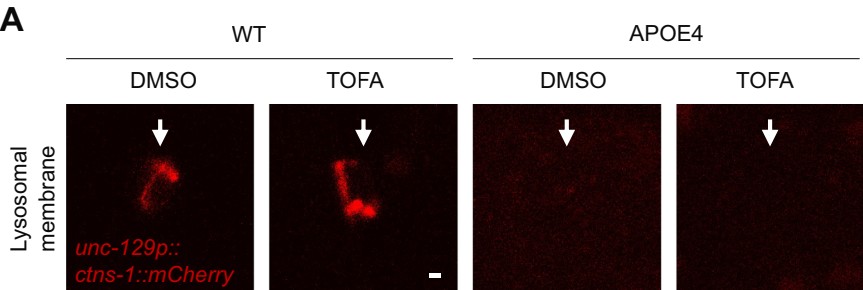

**Figure EV5. TOFA cannot rescue APOE4-induced neuronal lysosomal abnormalities in *C. elegans*.**

(A) Representative high magnification confocal microscopic images of neuronal specific lysosomal membrane reporter (white arrows) *cels56 [unc-129p::ctns-1::mCherry + nlp-21p::Venus + ttx-3p::RFP]*) at the YA stage, treated with DMSO or TOFA starting at the embryo stage. Scale bar: 1 μm.

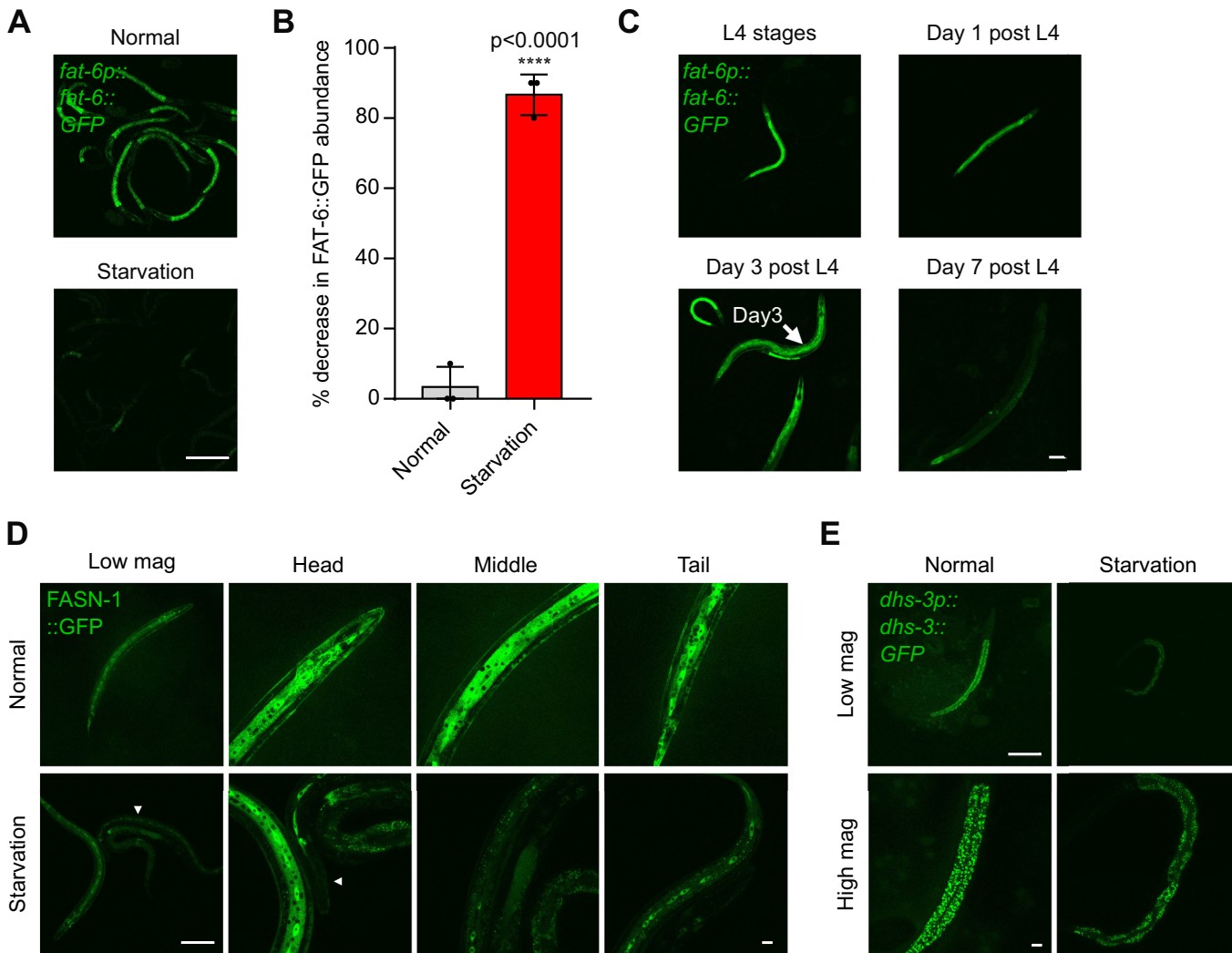

**Figure EV6. FAT-6::GFP downregulation during starvation and aging in *C. elegans*.**

(A) Representative confocal fluorescence images showing high-resolution Z-stack views of *fat-6p::fat-6::GFP* under 20 °C normal conditions or starvation of L1-L2 stages. Scale bars: 100 μm. (B) Quantification of the percentage of animals with *fat-6p::fat-6::GFP* fluorescence intensities downregulated under normal or starvation conditions. Data were presented as means ± S.D. *P* values calculated by unpaired two-tailed t-tests. ****P < 0.0001 (*n* > 25 animals per condition). (C) Representative confocal fluorescence images showing low-resolution Z-stack views of *fat-6p::fat-6::GFP* under 20 °C normal conditions for 0 day, 1 day, 3 days or 7 days post L4 stages. Scale bars: 100 μm. (D) Representative confocal fluorescence images showing low and high-resolution Z-stack views of FASN-1::GFP under 20 °C normal conditions or starvation (white arrows indicate positively changed worms). Scale bars: 100 μm or 10 μm (magnification). (E) Representative confocal fluorescence images showing low and high-resolution Z-stack views of DHS-3::GFP under 20 °C normal conditions or starvation. Scale bars: 100 μm or 10 μm (magnification).

