## [Peer Review File · EMBO Reports]

Conserved lipid metabolic reprogramming confers hypoxic and aging resilience

Wei Jiang, Goncalo Vale, Quentinn Pearce, Kaitlyn Kong, Wenbing Zhou, Jeffrey McDonald, James Cox, Neel Singhal, and Dengke Ma

Corresponding author(s): Dengke Ma (Dengke.Ma@ucsf.edu)

Review Timeline:	Transfer Date:	17th Oct 25
	Editorial Decision:	27th Oct 25
	Revision Received:	2nd Nov 25
	Accepted:	19th Nov 25

Editor: Yehu Moran

Transaction Report: This manuscript was transferred to EMBO reports following peer review at The EMBO Journal.

Dear Dengke,

Thank you for reaching out. Based on the EMBO Journal reviews and my own assessment, I consider the manuscript in principle suitable for EMBO Reports, and I would be happy to proceed toward acceptance without additional experiments, pending textual clarifications and revisions.

To move forward, please submit a revised manuscript together with a point-by-point response addressing all reviewer comments. In particular, please:

1. Explicitly discuss the omission of mitochondria and ROS in the current narrative (as raised by Referee #1), clarifying scope and limitations.

2. Address the mechanistic distinction noted by Referee #3, i.e., whether selective up-regulation of malonic acid would be expected to recapitulate the effects of triglyceride down-regulation, and why; indicate what is known vs. speculative.

3. Provide all essential methodological details requested by the referees (e.g., strain names, reagents, analysis parameters) so readers can understand and replicate both the experimental and computational results.

If you have questions about any of the above, I'm happy to advise. Please make sure to mention our correspondence in your cover letter and to address it specifically to me.

I look forward to handling your manuscript for EMBO Reports.

Best wishes,
Yehu

Yehu Moran, PhD
Professor, Head of the Department of Ecology, Evolution and Behavior
Alexander Silberman Institute of Life Sciences, Faculty of Science
The Hebrew University of Jerusalem
9190401 Jerusalem, Israel
Tel. Office: (+972)-2-658-5714 | Mobile: (+972)-50-549-9911
Email: yehu.moran@mail.huji.ac.il
Website: www.yehumoran.com

Referee #1:

The manuscript reports lipidomic and metabolomic profiling in arctic ground squirrel (AGS) neural stem cells compared to murine NSCs, identifying downregulation of triglyceride lipids and upregulation of malonic acid as a biosynthetic precursor. Inhibition of lipid biosynthesis enhanced hypoxic resilience in AGS NSCs, revealed conserved downregulation of lipid biosynthetic enzymes in *C. elegans*, and protected against hypoxia, APOE4-associated pathologies, and aging trajectories. These results suggest triglyceride modulation as an evolutionarily conserved resilience mechanism with implications for neural protection under stress conditions.

Strengths

- Novelty: This is the first work to link lipid metabolism, specifically triglyceride dynamics, with mitochondrial remodeling and whole-organism hypoxic resilience.
- Cross-species conservation across mammalian NSCs and *C. elegans* strengthens the mechanistic and evolutionary significance.
- Multi-modal approach combining lipidomics, functional perturbations, and model organisms provides breadth and mechanistic reach.
- The observation of a potential "sweet spot" in triglyceride regulation—where moderate downregulation confers protection but higher levels may contribute to vulnerability—is intriguing and conceptually important.
- Relevance extends to hypoxia/ischemia, APOE4-related neurodegeneration, and aging, increasing translational interest.

Weaknesses

Although the concept of a triglyceride "sweet spot" is compelling, the manuscript does not delve into what defines this critical threshold. Additional investigation into lipid peroxidation, ferroptosis, and their interplay with triglyceride content would strengthen the mechanistic conclusions.

The authors explicitly omit mitochondria and ROS for clarity, but these players are central to the proposed mechanism. Excluding them leaves the mechanistic pathway incomplete and weakens the translational framing.

While *C. elegans* findings are strong, mammalian validation beyond AGS NSCs is limited, leaving a gap for demonstrating clinical relevance.

The manuscript spans hypoxia, neurodegeneration, and aging, which could dilute the central message; connecting how tissue hypoxia or energy deficit may have a common theme within these three conditions would sharpen the impact.

This study makes a notable advance by identifying lipid metabolism as a critical determinant of hypoxic resilience, but further mechanistic depth regarding the triglyceride threshold and oxidative stress pathways would considerably enhance its strength.

Referee #2:

This study used metabolomic techniques to uncover metabolic adaptations that occur in the arctic ground squirrel to adapt to extreme environmental conditions, such as hypoxia. Compared to murine neuronal stem cells, the stem cells of the arctic ground squirrel accumulate reduced triglyceride levels under normal culture conditions, while other lipid levels are similar. In *C. elegans*, they showed that that fatty acid synthase and a Delta 9 fatty acid desaturase are down regulated when worms are exposed to hypoxia. By manipulating hypoxia inducible factor (HIF-1) and de novo fatty acid synthesis, they demonstrate in HEK293T cells that lipid droplets are reduced and this correlates with reduced membrane damage caused by hypoxia. In worms they reduced lipid synthesis using the *fasn-1* and *fat-5;fat-6* mutant strains, as well as a fatty acid synthesis inhibitor. In these studies they demonstrate increased resistance to hypoxia, as well as mitochondrial changes. Finally, they showed that the fatty acid synthesis inhibitor TOFA protects neurons from APOE damage and extends the lifespan of *C. elegans*. This work is important because it links fat synthesis with protection from hypoxia, neurodegeneration, and lifespan. Showing the evidence that changes in lipid synthesis is an important component of the stress resilience in two very different organisms is compelling. The paper is well-written, and the authors point out important limitations of their work. There are a few concerns that need to be addressed.

1. First Results section heading and text.

a. Is the term "downregulated" appropriate here, and similarly the wording of "remain unchanged"? The authors are doing a snapshot here, not an experiment where they change conditions and look for "upregulation or changes". My understanding is that they are talking about normal conditions. I think it is more appropriate to talk about abundance or composition, rather than regulation or changes.

b. Figure 1b. shows the relative abundance of lipid species. Is this relative to protein levels, or is it total lipids? This needs to be specified.

2. For the *C. elegans* mutant experiments. The strain *fasn-1(g14)* is reported to be

temperature sensitive and is best maintained at low temperature. The authors need to explain whether they manipulated the temperature to carry out their experiments.

3. The *C. elegans* lifespan experiments should be repeated without using FUDR. Also, all lifespan data should be reported in a table similar to the one shown in supplemental Table 22 from this paper: <https://doi.org/10.1038/s41586-019-1647-8>

Referee #3:

Jiang et al do a nice job of leveraging the unique attributes of the Arctic ground squirrel to discover a metabolic difference that may explain its resistance to damage in hypoxia, and testing if this generalizes to cultured cells (mouse) or whole animal (*C. elegans*). Specifically, their major discoveries are that 1) triglycerides are downregulated in neural stem cells derived from the Arctic ground squirrel in comparison to mouse, and 2) that enzymes that would produce triglycerides are downregulated in *C. elegans* exposed to hypoxia. A wide variety of approaches (HPLC, MS, genetics, pharmacology, imaging of cells and gene reporters) to substantiate the hypothesis that downregulation of triglycerides is a protective mechanism against stress.

Major issues:

The abstract implies that upregulation of malonic acid provides protection against hypoxia. It remains unaddressed whether selective upregulation of malonic acid would provide the same results as downregulation of triglycerides. This should be addressed at least in the Discussion and maybe tempered in the abstract without further experiments.

The type of triglycerides that impede protection against cellular and organismal insults when elevated was not studied. Moreover, the authors previously discovered that cholesterol played an important role in cellular damage in APOE4. The Discussion would benefit from a paragraph on relation between these triglycerides and cholesterol in protection.

Minor issues:

Line 99. Rather than saying "kit", concisely within the sentence explain the mechanism of detection for readers to compare.

Line 108. Would be nice to have graphic in Figure 1 to help readers quickly understand: "Malonic acid is a key intermediate in fatty acid biosynthesis, and its steady-state

accumulation is consistent with reduced triglyceride abundance in AGS, indicating downregulation of the fatty acid-to-triglyceride biosynthesis pathway". Or at least make it clearer in Fig2a.

Line 108. Was $p < 0.01$ significant after making many multiple comparisons in panel Fig 1F? If not, that's fine, but needs to be explained. If the correlation is underpowered due to so many comparisons, that could be used to justify further experimentation.

Line 181. Grammar/wording.

Line 232. Reword sentence to explain the control concisely.

Figure 2. Adjust or add to figure to help readers understand that fat-6 is a FAT, and what is being measured in panels f and g without unnecessary jargon. "Hif-1 target gene reporter" is confusing ; what target? Also, consider adding short diagram somewhere to help reader understand HIF-1, VHL-1 and drug VH-298.

Reword conclusion in Figure 4 title "...protects against hypoxia BY modulating mitochondrial fission" seems too strong because mitochondrial fission correlated with these conditions but fission was not experimentally tested as being required for hypoxia protection on its own.

Line 115. Grammar: are not is.

Line 340. Explain what the reporter signifies. Line 346. Explain what this is too for naïve readers. In Discussion, explain what both mean for hypoxia protection.

Figure 5 g. Indicate in representative photos the presence and absence of PVD neurites some way (e.g. solid and open triangles).

Figure 7. Help readers by showing where and how the few drugs in this study (Cerulenin, TOFA) target molecules or processes in this pathway. Or at least in Fig 2a.

Line 443. Remove the comma in "pathologies, and" to include the APOE4 result in the context of *C. elegans*, not on its own in another system.

A list of all *C. elegans* strains used should be included as supplemental, include strain names, genotypes, and sources.

UNIVERSITY OF CALIFORNIA, SAN FRANCISCO

BERKELEY · DAVIS · IRVINE · LOS ANGELES · RIVERSIDE · SAN DIEGO · SAN FRANCISCO · SANTA BARBARA

SANTA CRUZ · MERCED

Dear Prof. Moran,

Thank you very much for your thoughtful guidance and the opportunity to revise our manuscript entitled “**Conserved lipid metabolic reprogramming confers hypoxic and aging resilience**” (Jiang et al.) for *EMBO Reports*.

We have carefully revised the manuscript and fully addressed all comments from the *EMBO Journal* reviewers and your specific suggestions. In particular:

1. We have **explicitly discussed the omission of mitochondria and ROS**, clarifying both the scope and limitations of our study, as requested by Referee #1.
2. We have **addressed the mechanistic distinction raised by Referee #3**, elaborating on the relationship between malonic acid up-regulation and triglyceride down-regulation, and clarifying what is known versus speculative.
3. We have **included all essential methodological details** requested by the reviewers, including strain names, genotypes, reagents, and analysis parameters, to ensure reproducibility of both the experimental and computational results.

All reviewer comments have been addressed in detail in the accompanying **point-by-point response**. We appreciate your constructive feedback and guidance throughout this process.

We hope that the revised manuscript meets the standards of *EMBO Reports* and look forward to your favorable consideration for publication.

With best regards,

Dengke Ma

Dengke K. Ma, Ph.D.
Investigator, Cardiovascular Research Institute
Associate Professor, Department of Physiology
UCSF School of Medicine
555 Mission Bay Blvd South, SCVRB Room 252X
San Francisco, CA 94158-9001
Office phone: 415-502-3386
Email: Dengke.Ma@ucsf.edu

University of California
San Francisco

advancing health worldwide™

CVRI

Cardiovascular Research Institute

Responses to the reviewers' comments on Jiang et al., "Conserved lipid metabolic reprogramming confers hypoxic and aging resilience".

Referee #1:

The manuscript reports lipidomic and metabolomic profiling in arctic ground squirrel (AGS) neural stem cells compared to murine NSCs, identifying downregulation of triglyceride lipids and upregulation of malonic acid as a biosynthetic precursor. Inhibition of lipid biosynthesis enhanced hypoxic resilience in AGS NSCs, revealed conserved downregulation of lipid biosynthetic enzymes in C. elegans, and protected against hypoxia, APOE4-associated pathologies, and aging trajectories. These results suggest triglyceride modulation as an evolutionarily conserved resilience mechanism with implications for neural protection under stress conditions.

Strengths

- *Novelty: This is the first work to link lipid metabolism, specifically triglyceride dynamics, with mitochondrial remodeling and whole-organism hypoxic resilience.*
- *Cross-species conservation across mammalian NSCs and C. elegans strengthens the mechanistic and evolutionary significance.*
- *Multi-modal approach combining lipidomics, functional perturbations, and model organisms provides breadth and mechanistic reach.*
- *The observation of a potential "sweet spot" in triglyceride regulation—where moderate downregulation confers protection but higher levels may contribute to vulnerability—is intriguing and conceptually important.*
- *Relevance extends to hypoxia/ischemia, APOE4-related neurodegeneration, and aging, increasing translational interest.*

We thank the reviewer for the positive constructive comments.

Weaknesses

Although the concept of a triglyceride "sweet spot" is compelling, the manuscript does not delve into what defines this critical threshold. Additional investigation into lipid peroxidation, ferroptosis, and their interplay with triglyceride content would strengthen the mechanistic conclusions.

The authors explicitly omit mitochondria and ROS for clarity, but these players are central to the proposed mechanism. Excluding them leaves the mechanistic pathway incomplete and weakens the translational framing.

While C. elegans findings are strong, mammalian validation beyond AGS NSCs is limited, leaving a gap for demonstrating clinical relevance.

The manuscript spans hypoxia, neurodegeneration, and aging, which could dilute the central message; connecting how tissue hypoxia or energy deficit may have a common theme within these three conditions would sharpen the impact.

This study makes a notable advance by identifying lipid metabolism as a critical determinant of hypoxic resilience, but further mechanistic depth regarding the triglyceride threshold and oxidative stress pathways would considerably enhance its strength.

We thank the reviewer for these reasonable constructive comments. We plan to investigate all of the above, including lipid peroxidation, ferroptosis, and their interplay with triglycerides, as well as mammalian validation in AGS NSCs, in a follow-up study to further and better understand the underlying mechanisms.

Referee #2:

This study used metabolomic techniques to uncover metabolic adaptations that occur in the arctic ground squirrel to adapt to extreme environmental conditions, such as hypoxia. Compared to murine neuronal stem cells, the stem cells of the arctic ground squirrel accumulate reduced triglyceride levels under normal culture conditions, while other lipid levels are similar. In C. elegans, they showed that that fatty acid synthase and a Delta 9 fatty acid desaturase are down regulated when worms are exposed to hypoxia. By manipulating hypoxia inducible factor (HIF-1) and de novo fatty acid synthesis, they demonstrate in HEK293T cells that lipid droplets are reduced and this correlates with reduced membrane damage caused by hypoxia. In worms they reduced lipid synthesis using the fasn-1 and fat-5;fat-6 mutant strains, as well as a fatty acid synthesis inhibitor. In these studies they demonstrate increased resistance to hypoxia, as well as mitochondrial changes. Finally, they showed that the fatty acid synthesis inhibitor TOFA protects neurons from APOE damage and extends the lifespan of C. elegans. This work is important because it links fat synthesis with protection from hypoxia, neurodegeneration, and lifespan. Showing the evidence that changes in lipid synthesis is an important component of the stress resilience in two very different organisms is compelling. The paper is well-written, and the authors point out important limitations of their work. There are a few concerns that need to be addressed.

We thank the reviewer for the positive comments.

1. First Results section heading and text.

a. Is the term "downregulated" appropriate here, and similarly the wording of "remain unchanged"? The authors are doing a snapshot here, not an experiment where they change conditions and look for "upregulation or changes". My understanding is that they are talking about normal conditions. I think it is more appropriate to talk about abundance or composition, rather than regulation or changes.

We thank the reviewer for the suggestion, agree with the comments, and have revised accordingly in the Figure, Figures legends and the text.

b. Figure 1b. shows the relative abundance of lipid species. Is this relative to protein levels, or is it total lipids? This needs to be specified.

We thank the reviewer for the suggestion and have clarified that the relative abundance of lipid species is relative to protein levels accordingly.

2. For the C. elegans mutant experiments. The strain fasn-1(g14) is reported to be temperature sensitive and is best maintained at low temperature. The authors need to explain whether they manipulated the temperature to carry out their experiments.

We thank the reviewer for pointing this out, agree with the comments, and have revised accordingly in the text of figure legends and Methods.

3. The C. elegans lifespan experiments should be repeated without using FUDR. Also, all lifespan data should be reported in a table similar to the one shown in supplemental Table 22 from this paper: <https://doi.org/10.1038/s41586-019-1647-8>

We thank the reviewer for the suggestion and included lifespan data in a supplemental table 3.

Referee #3:

Jiang et al do a nice job of leveraging the unique attributes of the Artic ground squirrel to discover a metabolic difference that may explain its resistance to damage in hypoxia, and testing

if this generalizes to cultured cells (mouse) or whole animal (C. elegans). Specifically, their major discoveries are that 1) triglycerides are downregulated in neural stem cells derived from the Arctic ground squirrel in comparison to mouse, and 2) that enzymes that would produce triglycerides are downregulated in C. elegans exposed to hypoxia. A wide variety of approaches (HPLC, MS, genetics, pharmacology, imaging of cells and gene reporters) to substantiate the hypothesis that downregulation of triglycerides is a protective mechanism against stress.

We thank the reviewer for the positive comments.

Major issues:

The abstract implies that upregulation of malonic acid provides protection against hypoxia. It remains unaddressed whether selective upregulation of malonic acid would provide the same results as downregulation of triglycerides. This should be addressed at least in the Discussion and maybe tempered in the abstract without further experiments.

The type of triglycerides that impede protection against cellular and organismal insults when elevated was not studied. Moreover, the authors previously discovered that cholesterol played an important role in cellular damage in APOE4. The Discussion would benefit from a paragraph on relation between these triglycerides and cholesterol in protection.

We thank the reviewer for the suggestion and have revised accordingly in the Discussion text. It is an excellent point to raise an intriguing link to cholesterol given that triglyceride and cholesterol are major constituents of lipid droplets and lipoprotein particles and may thus jointly contribute the cellular damage caused by APOE4 or other oxidative stress conditions.

Minor issues:

1. Line 99. Rather than saying "kit", concisely within the sentence explain the mechanism of detection for readers to compare.

We thank the reviewer for pointing this out and have revised accordingly in the text to briefly clarify the mechanism of detection in the sentence.

2. Line 108. Would be nice to have graphic in Figure 1 to help readers quickly understand: "Malonic acid is a key intermediate in fatty acid biosynthesis, and its steady-state accumulation is consistent with reduced triglyceride abundance in AGS, indicating downregulation of the fatty acid-to-triglyceride biosynthesis pathway.". Or at least make it clearer in Fig2a.

We thank the reviewer for pointing this out and have included Figs. 1G-H accordingly.

3. Line 108. Was $p < 0.01$ significant after making many multiple comparisons in panel Fig 1F? If not, that's fine, but needs to be explained. If the correlation is underpowered due to so many comparisons, that could be used to justify further experimentation.

Yes, we thank the reviewer for pointing this out and have revised accordingly.

4. Line 181. Grammar/wording.

We thank the reviewer for pointing this out and have revised accordingly.

5. Line 232. Reword sentence to explain the control concisely.

We thank the reviewer for pointing this out and have revised accordingly.

6. Figure 2. Adjust or add to figure to help readers understand that fat-6 is a FAT, and what is being measured in panels f and g without unnecessary jargon. "Hif-1 target gene reporter" is confusing ; what target? Also, consider adding short diagram somewhere to help reader understand HIF-1, VHL-1 and drug VH-298.

We thank the reviewer for pointing this out and have revised the manuscript accordingly. Specifically, we clarified that *fat-6p::fat-6::GFP* is FAT-6::GFP, in Fig. 2C; we added information about the abundance of DHS-3::GFP and GFP measured in Figs. 2F and 2G in the text; we included the HIF-1 target gene reporter *nls470 (cysl-2p::Venus)* in Fig. 2G; and added a schematic diagram of HIF-1, VHL-1, and the drug VH-298 in Fig. EV2G.

7. Reword conclusion in Figure 4 title "...protects against hypoxia BY modulating mitochondrial fission" seems too strong because mitochondrial fission correlated with these conditions but fission was not experimentally tested as being required for hypoxia protection on its own.

We thank the reviewer for pointing this out and have revised accordingly.

8. Line 115. Grammar: are not is.

We thank the reviewer for pointing this out and have revised accordingly.

9. Line 340. Explain what the reporter signifies. Line 346. Explain what this is too for naïve readers. In Discussion, explain what both mean for hypoxia protection.

We thank the reviewer for pointing this out and have included information about the reporters *hsp-16p::GFP*, in which GFP is driven by the *heat shock protein-16* promoter, and *Q40::YFP*, which reflects length-dependent aggregation of polyglutamine (polyQ)-YFP driven by the *unc-54 promoter*. We have also revised the discussion accordingly.

10. Figure 5 g. Indicate in representative photos the presence and absence of PVD neurites some way (e.g. solid and open triangles).

We thank the reviewer for pointing this out and have revised accordingly to include solid and open triangles to indicate the presence and absence of PVD neurites, respectively.

11. Figure 7. Help readers by showing where and how the few drugs in this study (Cerulenin, TOFA) target molecules or processes in this pathway. Or at least in Fig 2a.

We thank the reviewer for pointing this out and have revised accordingly in Fig. EV 3A-B.

12. Line 443. Remove the comma in "pathologies, and" to include the APOE4 result in the context of *C. elegans*, not on its own in another system.

We thank the reviewer for pointing this out and have revised accordingly.

13. A list of all *C. elegans* strains used should be included as supplemental, include strain names, genotypes, and sources.

We thank the reviewer for pointing this out and have included all *C. elegans* strains as supplemental table 4.

Dear Dr. Ma

Thank you for the submission of your manuscript to our offices. EMBOR-2025-62983-T still has minor issues that I would like you to address before we can proceed with the official acceptance of your manuscript.

Our editorial assistance team noticed Figure re-use between - Figure 1E and Figure EV1A as well as between Figure 2H and Figure EV2C that is not listed in the figure legends. This must be addressed and clarified in the corresponding figure legends. Additionally, our team will follow with an email requesting from you relevant source data that should be uploaded by the time the paper is accepted and sent for production.

I look forward to seeing a new revised version of your manuscript as soon as possible.

Best wishes,
Yehu Moran
Academic Editor
EMBO Reports

The authors have addressed the editorial requests.

Dr. Dengke Ma
University of California, San Francisco
Cardiovascular Research Institute and Department of Physiology
555 Mission Bay Blvd South, SCVRB Room 252X
San Francisco, CA 94158
United States

Dear Dr. Ma,

I am pleased to inform you that your manuscript has been accepted for publication in EMBO reports. Your manuscript will be processed for publication by EMBO Press. It will be copy edited and you will receive page proofs prior to publication. Please note that you will be contacted by Springer Nature Author Services to complete licensing and payment information.

Yours sincerely,

Yehu Moran
Academic Editor
EMBO Reports
